TOPICAL REVIEW

# Neuromuscular mechanisms for the fast decline in rate of force development with muscle disuse – a narrative review

Luca Ruggiero [ID] and Markus Gruber [ID]

*Human Performance Research Centre, Department of Sports Science, University of Konstanz, Konstanz, Germany*

Handling Editors: Laura Bennet & Christoph Centner

The peer review history is available in the Supporting Information section of this article (https://doi.org/10.1113/JP285667#support-information-section).

**Abstract figure legend** Muscle unloading induces declines in muscle function, particularly in maximal and explosive strength. The decline in explosive strength (quantified as rate of force development, RFD) is greater than the decline in maximal strength (quantified as maximal force, $F_{max}$). This selective decline of explosive strength is the result of the interplay of neural, muscular, and tendinous adaptations with muscle disuse, which are hereby presented.

**Luca Ruggiero** is a researcher at the Human Performance Research Centre in the Department of Sport Science at the University of Konstanz (Germany). He completed his MSc at the University of Jyväskylä (Finland) and his PhD at the University of British Columbia (Kelowna, Canada). He is interested in the neuromechanics of movement and its optimisation, particularly in the contexts of explosive performance and fatigue. **Markus Gruber** is Chair in Training and Movement Science and Head of the Human Performance Research Centre in the Department of Sport Science at the University of Konstanz (Germany). He held research fellowships at the Universities Freiburg and Jyväskylä, before moving to the University Potsdam, and to the University Konstanz. His research focuses on human neuromuscular performance, with a particular emphasis on changes that occur across the lifespan and during prolonged periods of inactivity.

The Journal of Physiology

**Abstract**  The removal of skeletal muscle tension (unloading or disuse) is followed by many changes in the neuromuscular system, including muscle atrophy and loss of isometric maximal strength (measured by maximal force, $F_{max}$). Explosive strength, i.e. the ability to develop the highest force in the shortest possible time, to maximise rate of force development (RFD), is a fundamental neuromuscular capability, often more functionally relevant than maximal muscle strength. In the present review, we discuss data from studies that looked at the effect of muscle unloading on isometric maximal *versus* explosive strength. We present evidence that muscle unloading yields a greater decline in explosive relative to maximal strength. The longer the unloading duration, the smaller the difference between the decline in the two measures. Potential mechanisms that may explain the greater decline in measures of RFD relative to $F_{max}$ after unloading are higher recruitment thresholds and lower firing rates of motor units, slower twitch kinetics, impaired excitation-contraction coupling, and decreased tendon stiffness. Using a Hill-type force model, we showed that this ensemble of adaptations minimises the loss of force production at submaximal contraction intensities, at the expense of a disproportionately lower RFD. With regard to the high functional relevance of RFD on one hand, and the boosted detrimental effects of inactivity on RFD on the other hand, it seems crucial to implement specific exercises targeting explosive strength in populations that experience muscle disuse over a longer time.

(Received 21 June 2024; accepted after revision 27 September 2024; first published online 27 October 2024)

**Corresponding author** Luca Ruggiero: Human Performance Research Centre, Department of Sports Science, University of Konstanz, Mainaustraße 213, Konstanz 78464, Germany.     Email: luca.ruggiero@uni-konstanz.de

## Introduction

In the course of evolution, humans have developed a complex neuromuscular system to compromise capabilities such as maximal strength, steadiness, or fatigability, fundamental for the genus *Homo* (Marino et al., 2022). Another less visible but yet fundamental physical capacity of our neuromuscular system is to produce the highest force in the minimum time, referred to as rapid force capacity, or explosive strength (Maffiuletti et al., 2016; Rodriguez-Rosell et al., 2018). Explosive strength is critical for the performance of sport-specific as well as functional daily tasks or to avoid injuries or falls (Buckthorpe & Roi, 2018; Ema et al., 2016; Fleming et al., 1991; Maffiuletti et al., 2010).

Given the importance of explosive strength, several seminal studies have looked at underlying mechanisms behind training-specific improvements (e.g. Aagaard et al., 2002; Gruber et al., 2007; Van Cutsem et al., 1998) or lack thereof (e.g. Del Vecchio et al., 2022). Whereas on one side training can induce fast, considerable gains in explosive strength (Tillin et al., 2012a), considerable declines can be caused by acute conditions such as fatigue (Boccia et al., 2024; Buckthorpe et al., 2014) or chronic conditions such as biological ageing (Klass et al., 2008), orthopaedic limitations (e.g. knee or hip osteoarthritis; Maffiuletti et al., 2010; Suetta et al., 2007) and muscle mechanical unloading (e.g. lower limb suspension or bed rest; Monti et al., 2021; Rejc et al., 2018; Sarto et al., 2022).

Studying the deconditioning of muscle function, particularly of explosive strength, resulting from muscle mechanical unloading, is valuable for several reasons. First, for space travel, crews will be required to function autonomously for extended periods of time, under whole-body unweighting, without access to Earth facilities whilst being confronted with limited training capabilities (Williams & Turnock, 2011). Second, muscle disuse can occur as a result of injury, surgery, or frailty. Third, muscle unloading represents an experimental model to simulate biological ageing and physical inactivity (Biolo et al., 2003; Capri et al., 2023; Di Girolamo et al., 2021; Kehler et al., 2019; Sarto et al., 2023). Several paradigms have been used on humans to simulate muscle unloading on Earth as microgravitational analogues or to remove muscle tension: bed rest, dry immersion, limb immobilisation, and unilateral lower limb suspension (Adams et al., 2003; Gao et al., 2018; Pavy-Le Traon et al., 2007; Qaisar et al., 2020; Tomilovskaya et al., 2019). Given the fast (non-linear) atrophy and decline in muscle function with muscle unloading (Campbell et al., 2019; Marusic et al., 2021), several studies have focused on the counter-measures to avoid such declines (e.g. Clement et al., 2015; di Prampero, 2000; Gruber et al., 2019; Maffiuletti et al., 2019; Minetti et al., 2024; Ploutz-Snyder, 2016), and the time course of muscle function recovery with retraining (e.g. Hvid et al., 2010; Rejc et al., 2018; Sarto et al., 2022; Suetta et al., 2009). Fewer studies have focused on the decline in explosive strength, although explosive strength

has been considered to be more relevant than maximal strength to human overall performance (Lomborg et al., 2022; Maffiuletti et al., 2010; Orssatto et al., 2020).

The scope of this narrative review is to describe the effect of muscle mechanical unloading on isometric explosive strength. To address the specific mechanisms, we compare the decline in explosive strength with the decrease in isometric maximum strength. Regarding explosive strength, several excellent reviews have been published covering its neuromechanics and determinants in sports and clinical contexts and technical aspects of its measurement (Buckthorpe & Roi, 2018; Del Vecchio, 2023; Kozinc et al., 2022; Maffiuletti et al., 2016; Rodriguez-Rosell et al., 2018; Turpeinen et al., 2020). Moreover, acute conditions or measures such as fatigue (D'Emanuele et al., 2021), caffeine administration (Grgic & Mikulic, 2022), muscle heating (Rodrigues et al., 2022), or post-activation potentiation (Tillin & Bishop, 2009) were elaborated in detail. Also, seminal work has been published summarising the effect of muscle mechanical unloading on neuromuscular function (Campbell et al., 2019; Marusic et al., 2021; Narici & de Boer, 2011; Pavy-Le Traon et al., 2007; Qaisar et al., 2020). However, the effect and mechanisms of muscle unloading on explosive strength have not yet been the subject of in-depth analysis. This narrative review will focus on this latter issue, covering important aspects such as (1) the time course and magnitude of the decline in explosive relative to maximal strength; (2) the impact of unloading on the neuromechanical determinants of rapid force production, and (3) reasons for the pronounced decline in explosive relative to maximal strength.

### Why does explosive strength matter?

Whenever humans (or terrestrial animals) move, they apply force to change the linear velocity of their centre of mass or the angular velocity of their body segments. Considering them as rigid bodies, such change in velocity is proportional to the applied net force ($F$) or torque ($\tau$) according to the following relationships describing the production of impulse:

$$\int_{t1}^{t2} F \, \mathrm{d}t = m \, \Delta v \quad \int_{t1}^{t2} \tau \, \mathrm{d}t = I \, \Delta \omega,$$

where $t$ is the time, $m$ is the body mass, $I$ is the tensor of inertia, $v$ and $\omega$ are the linear and angular velocities, respectively. In practical terms, the greater the area under the force- or torque-time curve between the times $t1$ and $t2$ (the physical definition of impulse), the higher the change of linear or angular velocity for a given mass (the physical definition of momentum) (Aagaard et al., 2002). Thus, to achieve the highest possible change of velocity

during time-restricted movements, the highest amount of force in the minimum time must be developed (Aagaard et al., 2002; Rodriguez-Rosell et al., 2018). This physical capacity is referred to as explosive strength. The greater the explosive strength, the greater the acceleration in a propulsive movement (e.g. of the body centre of mass in jumping or of a thrown object), and the better the ability to control the centre of mass during decelerations or when counteracting a perturbation (e.g. to avoid a fall).

Explosive strength is a highly sought-after ability in the animal kingdom and in humans. In animals, explosive strength provides the sudden acceleration of the body centre of mass or body parts, generally to catch a prey or escape from predators (James et al., 2007). Given the high stakes, animals (depending on their size) have evolved highly specialised neuromechanical features like leg design or latch-mediated spring actuation to maximise explosive strength (Alexander, 1995; Patek, 2023), which robotics design is aspiring to achieve (Ilton et al., 2018) and learning from (Hawkes et al., 2022). Due to the high behavioural versatility required, the human neuromuscular system is not highly specialised for explosive strength. However, the capacity to produce the highest force in the minimum time is also of utmost importance for humans. During movement, the maximal force from a muscle group around a joint is hardly ever reached because by the time such maximal force is achieved, i.e. > 250 milliseconds in maximal voluntary isometric contractions (MVCs) across several muscle groups (Fig. 1), the outcome of a movement is typically already dictated. For this reason, it is very often critical to develop the highest force in the minimum time within such functional tasks.

Explosive strength from a muscle group around a joint is most commonly quantified through the rate of force development (RFD) in explosive rapid (or ballistic) efforts (e.g. Desmedt & Godaux, 1977a; Van Cutsem et al., 1998). Several studies have measured the rate of torque development instead of RFD, which is the rotational analogue. In the present review, we will address both measures as RFD. Likewise, when referring to the gold-standard measure of maximal strength, we will use MVC force ($F_{\max}$) to refer to both force and torque. To reduce the influence of muscle neuromechanical length- and velocity-dependent mechanisms (Hahn et al., 2017; Lieber & Ward, 2011), RFD is typically evaluated with isometric contractions. Exceptions include instances where explosive strength is purportedly studied in a dynamic scenario, e.g. eccentric or concentric (e.g. Monte et al., 2021; Tillin et al., 2012b, 2018), or in whole-body movements (e.g. in squat or countermovement jumps; Ruggiero et al., 2022). In addition, as neural activity immediately preceding a contraction inhibits RFD (Van Cutsem & Duchateau, 2005), explosive contractions are typically performed from a resting state.

As highlighted above, there is ample evidence demonstrating the relevance of explosive strength on functional outcomes. For example, indices of postural balance correlated with RFD from muscles of the lower limb (knee extensors: Izquierdo et al., 1999; Jakobsen et al., 2011; plantar flexors: Ema et al., 2016), and in the elderly the higher the RFD in a sit-to-stand task, the lower the risk of falling (Fleming et al., 1991). In people affected by unilateral long-term disuse such as unilateral hip osteoarthritis, RFD from the affected limb was considerably lower than in the unaffected side, and the inter-limb asymmetry of RFD was considerably greater than that of $F_{max}$ (Suetta et al., 2007). Six months after unilateral total knee arthroplasty, RFD was lower on the affected side, and inter-limb asymmetry in explosive but not maximal strength correlated with subjective knee function (Maffiuletti et al., 2010). Other examples pertain to sports and injury-risk related scenarios. Power athletes showed 100% greater RFD during knee extension but not more than 28% higher $F_{max}$ than control participants (Tillin et al., 2010), and RFD measured within the first 100 ms of explosive isometric squats was correlated with 20-m sprint performance, whereas $F_{max}$ was not (Tillin et al., 2013a). In isometric leg press, knee flexion and extension, as well as in countermovement and squat jumps, RFD is a pivotal measure to guide a safe return to sport after injuries in athletes such as after anterior cruciate ligament ruptures (Jordan et al., 2015, 2023). Consequently, testing explosive strength in single-joint or whole-body movements is considered a fundamental indicator for sport performance and health, and for returning to sport after musculoskeletal injuries (Buckthorpe & Roi, 2018).

## Explosive strength and power

Besides isometric conditions, explosive efforts have been also studied as the capacity to produce maximal power in dynamic explosive movements (e.g. Antonutto et al., 1999; Ferretti, 1997; Ferretti et al., 2001; Kramer et al., 2018; Rejc et al., 2015a, b, 2018). While the focus of the present review is on isometric explosive strength, it is necessary to note that in dynamic unconstrained (not iso-kinetic) movements explosive power and strength are not equivalent but are still physically related (Aagaard et al., 2002; Minetti, 2002; Zamparo et al., 2002).

To explain this, let's consider the mechanism represented in Fig. 2A: a cube of mass $m = 1$ kg, connected to a massless linear actuator (a hollow and a cylinder). The actuator starts from stationary conditions, pulling up the cube with a force directed upwards with constant RFD of 60 N s$^{-1}$. Gravitational acceleration $g$ was set to 9.81 m s$^{-2}$. The system was solved analytically, and numerically verified with Simscape Multibody (MATLAB v2023b; MathWorks, Natick, MA, USA).

Power ($\dot{W}$; in Watts) is the rate at which mechanical work is performed, and it corresponds to the product of the force ($F$; in N) and velocity ($v$; in m s$^{-1}$) as functions of time:

$$\dot{W}(t) = F(t)\, v(t).$$

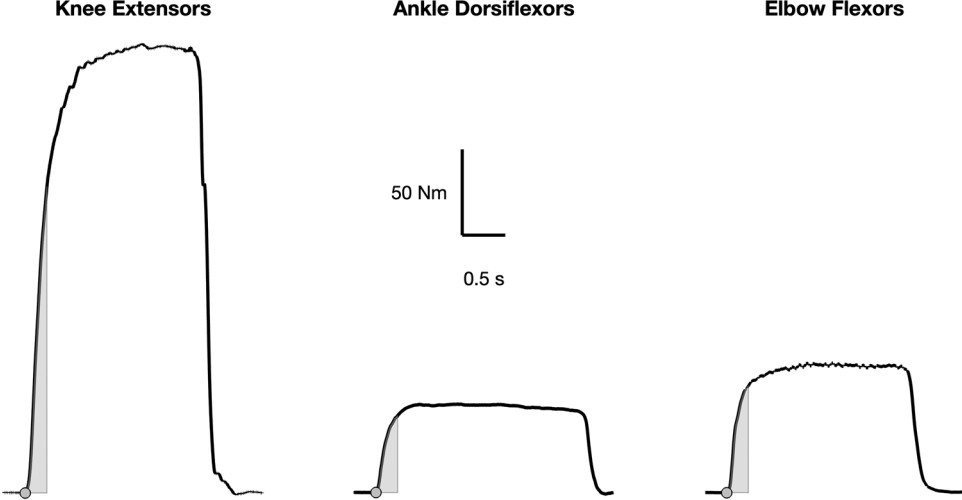

**Figure 1. Example of maximal voluntary isometric contractions (MVCs) from knee extensors, ankle dorsiflexors, and elbow flexors**
Data for knee extensors are from Ruggiero, Hoiland, et al. (2018), data for ankle dorsiflexors were collected at the University of Konstanz (authors' unpublished data), and data for elbow flexors are from Ruggiero and McNeil (2023). Grey filled circles represent the onset of force (manually determined; Tillin et al., 2013b). Grey areas represent the impulse produced during the first 250 ms of the MVCs. Only a fraction of maximal force is produced in the first 250 ms from force onset. Therefore, during functional tasks, the capacity to develop the greatest force in the minimum time (i.e. explosive strength) is critical.

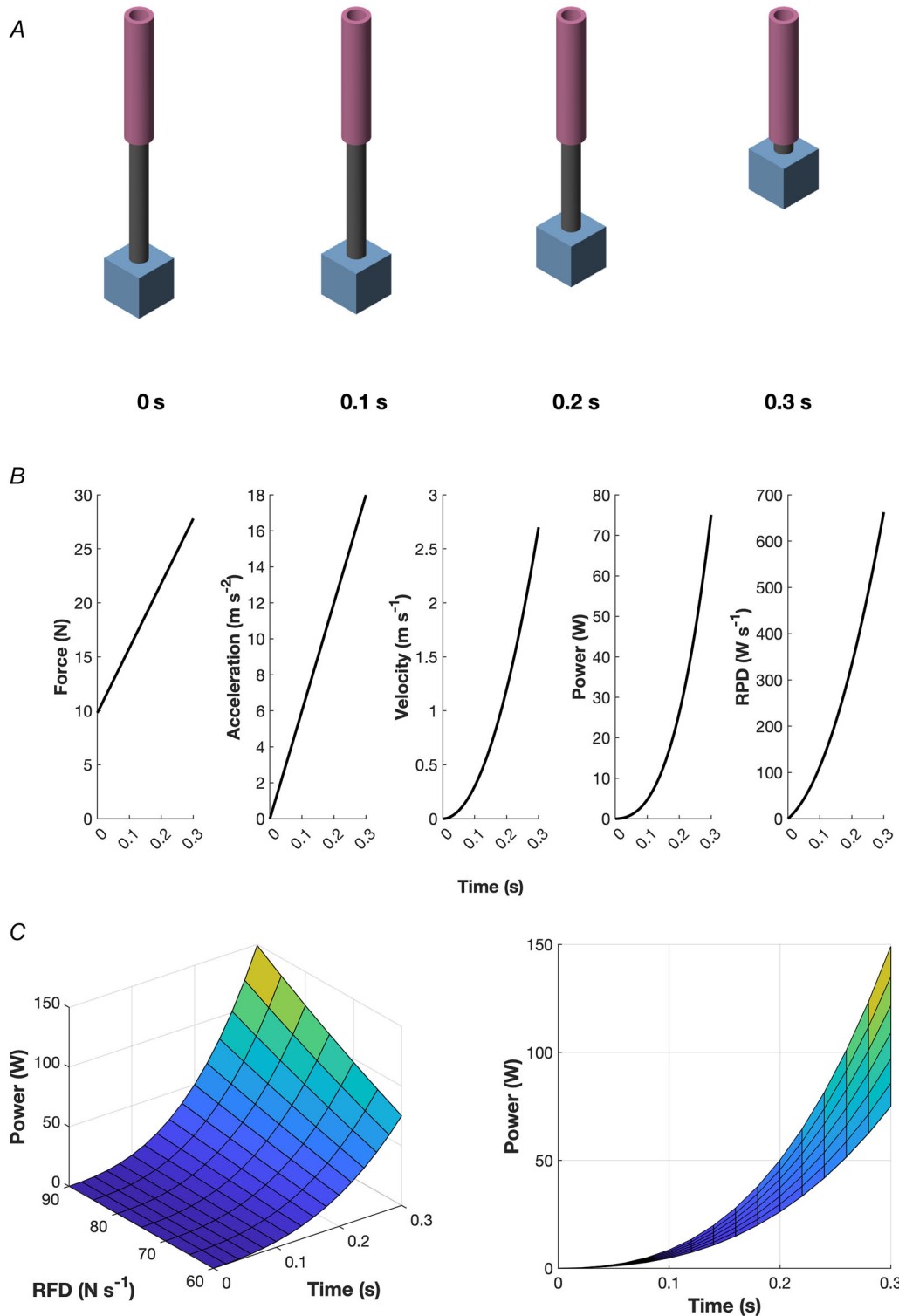

**Figure 2. The dynamics of a mechanical actuator pulling up a mass from a stationary condition with a constant rate of force development (RFD)**

*A*, the system at the start of the simulation and after 0.1, 0.2 and 0.3 s. The actuator is composed of one hollow and one full massless cylinder, pulling up a cube of mass 1 kg with a force directed upwards. The system starts from stationary conditions. The gravitational acceleration has been set to 9.81 m s$^{-2}$. The upward force starts from 9.81 N and increases with a constant RFD of 60 N s$^{-1}$. Images are from the simulation conducted in Simscape Multibody. *B*, the resulting force (in N), acceleration (in m s$^{-2}$), velocity (in m s$^{-1}$), power ($\dot{W}(t)$, in W), and rate

of power development (RPD or $\ddot{W}(t)$; in W s$^{-1}$). Power and rate of power development are physically related to RFD according to the equations $\dot{W}(t) = (\frac{RFD\,t}{m} + g)\frac{RFD\,t^2}{2}$ and $\ddot{W}(t) = \frac{RFD}{2\,m}(3\,RFD\,t^2 + 2\,m\,g\,t)$, where $t$ is the elapsed time (s), $m$ is the mass of the cube (kg) and $g$ is the gravitational acceleration (m s$^{-2}$). See main text for the analytical explanation. C, several $\dot{W}(t)$ functions with RFD varying from 60 to 90 N s$^{-1}$ in steps of 5, and their projections on the power-time plane.

Force is given by:

$$F(t) = ma(t) + mg,$$

where $a(t)$ is the acceleration function (in m s$^{-2}$). The system has been defined with a constant RFD, starting from stationary conditions ($F = mg$). Thus:

$$F(t) = RFD\,t + mg,$$

where $t$ is the elapsed time (s). The velocity function (in m s$^{-1}$) can be derived as:

$$v(t) = \int_0^t a(t)\,\mathrm{d}t.$$

Rearranging the equations above:

$$v(t) = \int_0^t a(t)\,\mathrm{d}t = \int_0^t \left(\frac{F(t)}{m} - g\right)\mathrm{d}t.$$

And substituting the equation for force:

$$v(t) = \int_0^t \left(\frac{RFD\,t + mg}{m} - g\right)\mathrm{d}t = \int_0^t \left(\frac{RFD\,t}{m}\right)$$

$$\mathrm{d}t = \frac{RFD\,t^2}{2\,m}.$$

The power of the system as function of time can then be calculated as:

$$\dot{W}(t) = F(t)\,v(t) = \left(RFD\,t + mg\right)\frac{RFD\,t^2}{2\,m}$$

$$= \left(\frac{RFD\,t}{m} + g\right)\frac{RFD\,t^2}{2}.$$

Rate of power development ($\ddot{W}(t)$, or RPD), an important metric that represents performance in unconstrained movements (Jakobsen et al., 2012; Ruggiero et al., 2022), can be derived as the time derivative of the power function:

$$\ddot{W}(t) = \frac{RFD}{2\,m}\left(3\,RFD\,t^2 + 2\,m\,g\,t\right).$$

Such analytical solutions are depicted in Fig. 2*B* (for $m = 1$ kg; massless linear actuator; $g = 9.81$ m s$^{-2}$; RFD = 60 N s$^{-1}$). The last two equations formally show that power and rate of power development as functions of time ($\dot{W}(t)$ and $\ddot{W}(t)$, respectively) during the modelled movement both depend on RFD with a quadratic relationship. Variations in RFD directly affect the power outcome during the movement as represented in Fig. 2*C*, where several $\dot{W}(t)$ functions and their

projections on the power-time plane are depicted with RFD from 60 to 90 N s$^{-1}$ in steps of 5.

The system modelled in Fig. 2*A* is characterised by initial stationary conditions and a linear movement with a constant RFD. While conceptual conclusions regarding the relationship between RFD, power, and RPD are valid, there are a few caveats: (1) in explosive contractions, RFD is not constant, and it changes based on time from force onset (e.g. Folland et al., 2014, see Fig. 2*E* and *F* of Del Vecchio, Negro et al., 2019); (2) RFD (from a muscle or a muscle group) is affected by muscle length- and velocity-dependent mechanisms (Hahn et al., 2017; Lieber & Ward, 2011); (3) regardless of the type of contraction (isometric or dynamic), RFD is affected by changes in muscle fascicle pennation angle and shifts in the muscle belly gear ratio (Monte, 2020; Monte & Zignoli, 2021; Monte et al., 2021; Van Hooren et al., 2024; see also section 'Determinants of explosive strength' below). The equations above refer to a linear movement, but the same conclusions are valid for a rotational analogue. Regardless of the model (linear or rotational), the general principle remains: in unconstrained movements (not isokinetic), RFD, power and RPD are tightly physically coupled (Aagaard et al., 2002; Minetti, 2002; Zamparo et al., 2002).

## Determinants of explosive strength

Explosive contractions are characterised by the development of the highest force in the minimum time (<200 ms). As earlier reported, to reduce the influence of muscle-length and contraction-velocity dependent mechanisms (Hahn et al., 2017; Lieber & Ward, 2011), explosive strength has been typically evaluated during isometric contractions. Isometric explosive strength depends primarily on how fast motor units (MUs) are recruited at the beginning of the ballistic contraction, although other factors (e.g. instantaneous MU firing rate and doublets; muscular and tendon mechanical properties) to a lower extent, influence force output (Del Vecchio, 2023; Dideriksen et al., 2020; Duchateau & Baudry, 2014; Maffiuletti et al., 2016; Monte et al., 2021).

The first work by Desmedt and Godaux (1997a, b, c) using needle electromyography (EMG) highlighted that explosive contractions are characterised by a high initial recruitment rate of MUs, high MU firing rates, and reduced recruitment threshold, relative to MVCs. Important studies have then reinforced the tight dependence of RFD on MU firing rate (Duchateau &

Baudry, 2014; Van Cutsem et al., 1998; Van Cutsem & Duchateau, 2005), as well as the importance of the pre-contraction silent period of the MUs to maximise their initial firing rate, and in turn increase RFD (Aoki et al., 2002; Moritani & Shibata, 1994; Tsukahara et al., 1995; Van Cutsem & Duchateau, 2005; Walter, 1989).

In ballistic contractions, most of the variance in the RFD calculated 0–50 ms from contraction onset was found to be explained by the rate of EMG rise, whereas RFD in later windows (>50 ms) depended substantially on maximal involuntary and voluntary muscle force production capabilities (Cossich & Maffiuletti, 2020; D'Emanuele et al., 2022; Folland et al., 2014). While it is practical to subdivide ballistic contractions and analyse RFD in discrete time windows, explosive strength should be considered a neuromechanical continuum, where the neural and muscular components behave like a single entity (Del Vecchio, 2023).

A combination of high-density EMG and modelling studies have identified that the capacity of the neuromuscular system to produce force at a maximal rate is mostly dictated by the recruitment rate of MUs, followed by initial instantaneous firing rate, likelihood of doublet discharges, and twitch contractile properties (~fourfold, fivefold, and sixfold less influential than MU recruitment rate, respectively; Del Vecchio, Negro et al., 2019; Dideriksen et al., 2020). Surprisingly, the synchronisation of MUs because of the common synaptic input to MUs (Farina & Negro, 2015) and the high discharge rate of MUs (de la Rocha et al., 2007), may not additionally influence RFD in explosive contractions (Del Vecchio, Falla, et al., 2019).

Within the neuromechanical continuum, muscle and tendon properties may not be overlooked. Indeed, the greater the tendon-aponeurosis or tendon stiffness of a muscle-tendon complex, the greater the RFD in isometric contractions (Bojsen-Møller et al., 2005; Mayfield, Cresswell & Lichtwark, 2016; Monte, 2020; Monte & Zignoli, 2021; Wang et al., 2012), suggesting that such property of the connective tissue increases the ability to transmit force more directly, a finding that is common in nature (e.g. in muscle fibres of anurans; Mayfield, Launikonis, et al., 2016). Despite this, inconsistencies where connective tissue stiffness did not independently affect RFD have been reported (Hannah & Folland, 2015; Massey et al., 2017). The stiffness of the muscle tissue can also be positively related to force production in explosive contraction with a higher muscle stiffness correlated to a greater peak RFD (Monte & Zignoli, 2021). In addition, the greater the muscle belly gearing (i.e. the ratio between muscle belly velocity and muscle fibre velocity; Azizi et al., 2008; estimated as belly segment gear; Pinto et al., 2023, which is highly correlated with muscle stiffness; Monte & Zignoli, 2021), the greater the RFD in explosive efforts (Monte, 2020; Monte & Zignoli, 2021; Monte et al.,

2021; Van Hooren et al., 2024). Indeed, a higher muscle belly gearing allows muscles to produce greater forces when contracting faster or at higher velocities for the same effective muscle fascicle contraction velocity (Dick & Wakeling, 2017; Eng et al., 2018).

Moving from isometric to dynamic contractions, further factors must be taken into account. In isokinetic fast concentric contractions of the knee extensors, the higher the contraction speed, the greater the RFD relative to maximal isokinetic force (Tillin et al., 2021). In such fast conditions, the force- and power-velocity relationships of muscle fascicles (from vastus lateralis) impose a limit to explosive strength (Monte et al., 2021; Werkhausen et al., 2022), which is circumvented by the muscle belly (segment) gearing through increased muscle thickness and pennation angle (Monte et al., 2021). When fast eccentric contractions of the knee extensors are performed, the RFD is lower than in concentric contractions for matched angular position, in contrast to what is expected from the classical torque-angle-angular velocity relationship (eccentric torque > concentric torque; Tillin et al., 2012b). When normalised to maximal force, the higher the acceleration and contraction speed, the lower the RFD in eccentric *vs.* concentric contractions (Tillin et al., 2018). Such discrepancy between eccentric and concentric contractions most likely comes from neural inhibition of force production in the former modality, typically present at high contraction speed (Aagaard, 2018; Duchateau & Enoka, 2016), and evidenced in explosive fast contractions by surface EMG and the ratio between voluntary and evoked eccentric RFD (Tillin et al., 2012b).

This review will consider only isometric conditions. However, when relevant, reference will be made to dynamic explosive movements.

## Paradigms of muscle mechanical unloading

The detection and response to external load is critical for the skeletal muscle tissue, whose mechanical, metabolic, and endocrine functions depend on muscle tension (Hoffmann & Weigert, 2017; Lieber et al., 2017). In situations such as spaceflight, illness, recovery from acute injury, or the presence of chronic conditions such as osteoarthritis or ageing, the reduction of local muscle tension induces notable remodelling in the neuromuscular system. For this reason, experimental models have been used to simulate such drastic reductions in muscle mechanical load. The most frequently used ground-based models in humans are bed rest, dry immersion, limb immobilisation and unloading (Adams et al., 2003; Campbell et al., 2019; Gao et al., 2018; Pavy-Le Traon et al., 2007; Qaisar et al., 2020). Each method of muscle mechanical unloading has its pros and cons.

**Table 1. Details of the studies that report the effect of muscle mechanical unloading on maximal and explosive isometric voluntary force**

| Study | Unloading paradigm | Length (days) | Muscle group | Isometric method (joint angle) | Measure | |
|---|---|---|---|---|---|---|
| | | | | | Explosive strength | Muscle group CSA* |
| Hvid et al. (2014) | IM | 4 | KE | Knee extension (110°) | RFD: 0–100 ms Onset: 3% $F_{max}$ | Muscle fibres CSA |
| Monti et al. (2021) | BR | 10 | KE | Knee extension (90°) | RFD: 0%–63% $F_{max}$ Onset: not specified | QF CSA |
| Sarto et al. (2022) | ULLS | 10 | KE | Knee extension (90°) | RFD: 0%–63% $F_{max}$ Onset: not specified | QF CSA |
| Bamman et al. (1998) | BR | 14 | KE | Knee extension (120°) | RFD: 10%–60% $F_{max}$ | Muscle fibres CSA |
| Hvid et al. (2010); Suetta et al. (2009) | IM | 14 | KE | Knee extension (110°) | RFD: 0%–66% $F_{max}$ Onset: 3% $F_{max}$ | QF volume |
| Kubo et al. (2000) | BR | 20 | KE | Knee extension (100°) | RFD: 10%–60% $F_{max}$ | QF CSA |
| Horstman et al. (2012) | ULLS | 21 | PF and KE | Plantar flexion (-15°); Knee extension (120°) | RFD: maximum of force 1st time derivative | None |
| De Boer et al. (2007) | ULLS | 23 | KE | Knee extension (100°) | RFD: 0–100 ms; Onset: 2 Nm above baseline | QF CSA |
| Valdes et al. (2020) | IM | 28 | EF | Elbow flexion (90°) | RFD: 0–100 ms; Onset: 8 N above baseline | Arm circumference |
| Mulder et al. (2006, 2008) | BR | 56 | KE | Knee extension (110-120°), supine position | Impulse: 0–40 ms; Onset: > 3 SDs baseline force | QF CSA |
| Mulder et al. (2009) | BR | 56 | PF and KE | Plantar flexion (0°); Knee extension (120°) | RFD: maximum of force 1st time derivative | QF and TS CSA |
| Kramer et al. (2021) | BR | 60 | PF and KE | Plantar flexion (0°); Knee extension (120°) | RFD: maximum of force 1st time derivative | None |
| Alkner and Tesch (2004); Alkner et al. (2016) | BR | 90 | LL | Supine squat with knee extended (90°) | RFD: 100–200 ms; Onset: not specified | QF and TS volume |

Characteristics of the studies include the unloading paradigm used and its duration, the muscle group examined, the joint position for isometric assessments, the measure of RFD derived from the study, and the retrieved measure of muscle group CSA (or estimate). For details regarding the retrieval of data, see Supporting information, item A1. BR: bed rest; CSA: cross-sectional area; EF: elbow flexors; $F_{max}$: isometric maximal voluntary contraction force or torque; IM: limb immobilisation; KE: knee extensors; LL: lower limb; PF: plantar flexors; QF: quadriceps femoris; RFD: isometric rate of force or torque development; TS: triceps surae; ULLS: unilateral lower limb suspension.
*If muscle group CSA was not available, the following measures were considered in order: muscle volume, mean muscle fibres CSA, limb circumference.

Briefly, bed rest and dry immersion consist of lying down on a bed for days or weeks (most often combined with head-down tilt) or in a water-filled tank (through water-proof cloth, inducing effective uniform buoyant forces), respectively (Watenpaugh, 2016). These two models allow reproduction of many of the cardiovascular and skeletal muscle-related aspects of adaptations to microgravity in spaceflight (Adams et al., 2003; Convertino, 1996; Qaisar et al., 2020; Tomilovskaya et al., 2019; see Table 1 in Pavy-Le Traon et al., 2007, adapted from Nicogossian et al., 1994; see Table 2 in Qaisar et al., 2020), representing at the moment the best models on Earth to reproduce the full spectrum of changes with exposure to microgravity (Pavy-Le Traon et al., 2007; Tomilovskaya et al., 2019). Limb immobilisation and unloading are more suitable to understand the

**Table 2. Percentage declines of $F_{max}$ and RFD relative to baseline after the unloading protocol, and their difference**

| Study | Unloading paradigm | Length (days) | Muscle group | Δ % (Relative to baseline) | | Difference in Δ % ($F_{max}$ − RFD) |
|---|---|---|---|---|---|---|
| | | | | $F_{max}$ | RFD | |
| Hvid et al. (2014) | IM | 4 | KE | −9.9 | −14.8 | 4.9 |
| Monti et al. (2021) | BR | 10 | KE | −13.5 | −37.0 | 23.5 |
| Sarto et al. (2022) | ULLS | 10 | KE | −29.5 | −54.4 | 24.9 |
| Bamman et al. (1998) | BR | 14 | KE | −14.5 | −54.4 | 39.9 |
| Hvid et al. (2010); Suetta et al. (2009) | IM | 14 | KE | −15.8 | −18 | 2.2 |
| Kubo et al. (2000) | BR | 20 | KE | −19.2 | −47.1 | 27.9 |
| Horstman et al. (2012) | ULLS | 21 | PF and KE* | −16.5 | −16.2 | −0.3 |
| De Boer et al. (2007) | ULLS | 23 | KE | −20.3 | −37.8 | 17.5 |
| Valdes et al. (2020) | IM | 28 | EF | −21.6 | −25.6 | 4.0 |
| Mulder et al. (2006, 2008) | BR | 56 | KE | −21.5 | −21.5 | 0 |
| Mulder et al. (2009) | BR | 56 | PF and KE* | −15.3 | −20.5 | 5.2 |
| Kramer et al. (2021) | BR | 60 | PF and KE* | −40.9 | −34.0 | −6.9 |
| Alkner and Tesch (2004); Alkner et al. (2016) | BR | 90 | LL | −45.0 | −45.3 | 0.3 |

BR: bed rest; CSA: cross-sectional area; EF: elbow flexors; $F_{max}$: isometric maximal voluntary contraction force or torque; IM: limb immobilisation; KE: knee extensors; LL: lower limb; PF: plantar flexors; RFD: isometric rate of force or torque development; ULLS: unilateral lower limb suspension. For details on how measures were retrieved from the individual studies, see Supporting information, item A1.

*The percentage decline in $F_{max}$ and RFD was averaged between the two muscle groups considered.

mechanisms that underlie unloading-induced atrophy of skeletal muscles and the functional consequences. The first model (immobilisation) is characterised by fixation of a joint (most commonly the knee-joint; Campbell et al., 2019) in a typically slightly flexed position (e.g. Deschenes et al., 2008), whereas the second model (unloading) has been most commonly achieved by using a support strap to suspend one lower limb, preventing weightbearing, while wearing a shoe with a high outsole platform on the contra-lateral limb (e.g. Berg et al., 1991; Sarto et al., 2022). The effects of these two ground-based models are limited to the muscles of the immobilised limb (although systemic effects are still present; Adams et al., 2003). Whether selecting bed rest, dry immersion, limb immobilisation or unloading therefore depends on the specific aim of the study.

A very attractive approach to simulate reduction of muscle external loading and sedentarism is step reduction (for a recent review, see Sarto et al., 2023). However, this model falls outside of the scope of the present review, as it represents a milder muscle unloading stimulus compared to bed rest, dry immersion, limb immobilisation or unloading, being mainly used by researchers interested in sedentarism (Sarto et al., 2023). For the same reason the present review does not consider disuse-related muscle unloading resulting from long-term orthopedic

limitations such as knee or hip osteoarthritis (Loureiro et al., 2018; Maffiuletti et al., 2010; Suetta et al., 2007), that typically represent a weaker unloading stimulus than the ground-based models described above, and may be impacted by other factors than muscle mechanical unloading only, e.g. decreasing overall activity levels due to pain.

### Effect of muscle mechanical unloading on muscle maximal and explosive voluntary force

The studies examining (concurrently) the effect of muscle mechanical unloading on isometric maximal and explosive voluntary force are reported in Table 1, with details of the unloading paradigm used and its duration, the muscle group examined, the joint angular position for isometric evaluation, and the metrics defining RFD and muscle group CSA. Specific details on the procedures used to extract data from individual studies are reported in the Supporting information, item A1. Table 2 reports the percentage decline (as a percentage of baseline) from the unloading protocol of the absolute values of $F_{max}$ and RFD, and their difference. Table 3 reports the decrease of muscle group CSA (or estimates) from the unloading protocol and of the same measures reported in Table 2 but there

**Table 3. Percentage declines of muscle group CSA, $F_{max}$ and RFD normalised to muscle group CSA (relative to baseline) after the unloading protocol, and their difference**

| Study | Unloading paradigm | Length (days) | Muscle group | Δ % (Relative to baseline) | | | Difference in Δ % ($F_{max}$ - RFD) |
| --- | --- | --- | --- | --- | --- | --- | --- |
| | | | | Muscle group CSA | Normalised $F_{max}$ | Normalised RFD | |
| Hvid et al. (2014) | IM | 4 | KE | −10.3 | −1.7 | −7.3 | 5.6 |
| Monti et al. (2021) | BR | 10 | KE | −3.3 | −10.5 | −34.8 | 24.3 |
| Sarto et al. (2022) | ULLS | 10 | KE | −4.6 | −26 | −52.2 | 26.2 |
| Bamman et al. (1998) | BR | 14 | KE | −16.8 | −1.2 | −47.3 | 46.1 |
| Hvid et al. (2010); Suetta et al. (2009) | IM | 14 | KE | −8.9 | −8.4 | −20.7 | 12.3 |
| Kubo et al. (2000) | BR | 20 | KE | −7.5 | −12.6 | −42.8 | 30.2 |
| De Boer et al. (2007) | ULLS | 23 | KE | −10 | −11.5 | −31 | 19.5 |
| Valdes et al. (2020) | IM | 28 | EF | −9.8 | −13.1 | −16.2 | 3.1 |
| Mulder et al. (2006, 2008) | BR | 56 | KE | −14.2 | −1.3 | −7.3 | 6.0 |
| Mulder et al. (2009) | BR | 56 | PF and KE* | −13.7 | −4.7 | −5.4 | 0.7 |
| Alkner and Tesch (2004); Alkner et al. (2016) | BR | 90 | LL | −22 | −29.6 | −31 | 1.4 |

BR: bed rest; CSA: cross-sectional area; EF: elbow flexors; $F_{max}$: isometric maximal voluntary contraction force or torque; IM: limb immobilisation; KE: knee extensors; LL: lower limb; PF: plantar flexors; RFD: isometric rate of force or torque development; ULLS: unilateral lower limb suspension. For details on how measures were retrieved from the individual studies, see Supporting information, item A1. Values from Horstman et al. (2012) and Kramer et al. (2021) were removed as no measures of CSA could be retrieved.
*The percentage decline in $F_{max}$ and RFD was averaged between the two muscle groups considered.

normalised to muscle group CSA. Studies not reporting measures or estimates of CSA (i.e. Horstman et al., 2012 and Kramer et al., 2021) were not included in Table 3. The studies conclusively show that after onset of muscle mechanical unloading explosive strength decays faster than maximal strength, and the longer the duration of the unloading protocol, the smaller the difference between the decline in the two measures (Fig. 3D and E). This trend emerges regardless of whether data are normalised to muscle group CSA or considered in absolute values, even though the normalisation procedure may numerically affect the results (see Supporting information, item A2).

Muscle unloading results in a progressive decline in muscle group CSA and $F_{max}$ (Fig. 3A). Such decline with time has been previously modelled applying logarithmic or exponential relationships (e.g. Fig. 1 in Marusic et al., 2021; Fig. 1 in Ferretti, 1997). From the studies reported in Table 1, the percentage decline (relative to baseline) of muscle group CSA was modelled (using MATLAB v2023b; MathWorks, Natick, MA, USA) applying a logarithmic equation $y = -3.57 \times \ln(x + 1)$ ($R^2$: 0.49; 95% confidence bounds: $[-4.40, -2.74]$), where $x$ represents the duration (in days) of the unloading protocol. Similarly, the decline in $F_{max}$ was modelled as $y = -6.96 \times \ln(x + 1)$ ($R^2$: 0.42; 95% confidence bounds:

$[-8.46, -5.47]$). The logarithmic function was imposed the passage from the point (0,0), for no decline in muscle group CSA and $F_{max}$ is expected with no muscle unloading. As such, the argument of the natural logarithm was $(x + 1)$. The logarithmic function was preferred to an exponential (decaying) function, as a horizontal asymptote in our data could not be determined (necessary when exponential functions of the type $y = e^{-\frac{x}{\tau}}$ are modelled, e.g. as in Fig. 1 in Ferretti, 1997). The declines of muscle group CSA and $F_{max}$ in the studies herein considered are very similar (dashed lines in Fig. 3A) to those reported in a recent comprehensive review (Marusic et al., 2021. Note that six studies have been considered both in the present review and in Marusic et al., 2021). As previously highlighted (e.g. see the review of Marusic et al., 2021, considering bed rest protocols, or the review of Campbell et al., 2019, considering limb immobilisation or lower limb suspension), the atrophy and the loss of $F_{max}$ are non-uniform with time, and the magnitude of the latter is greater than the former.

Muscle unloading results in a considerable decrease of explosive strength, which outweighs the decline in maximal strength. Figure 3B reports the percentage change of absolute $F_{max}$ and RFD as function of the unloading protocols duration (data from Table 2), whereas

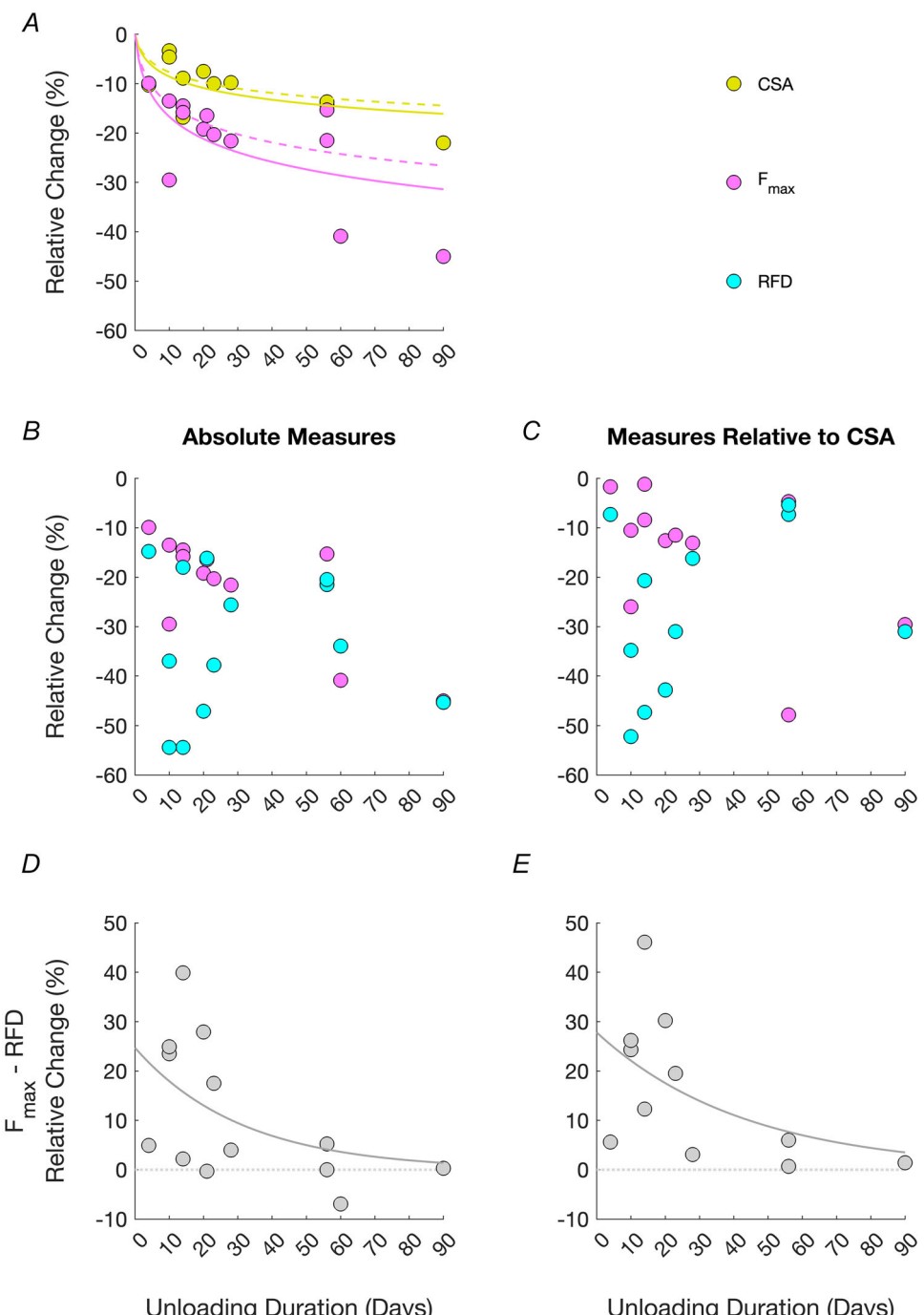

**Figure 3. Percentage changes (relative to baseline) of muscle group cross-sectional area (CSA), isometric maximal and explosive strength ($F_{max}$ and RFD, respectively), and difference between changes in $F_{max}$ and RFD as a function of the duration of muscle mechanical unloading**

Data points represent results from studies reported in Tables 2 ($F_{max}$ in Fig. 3*A*, and data in Fig. 3*B* and *D*) and 3 (CSA in Fig. 3*A*, and data in Fig. 3*C* and *E*). *A*, percentage change (relative to baseline) of muscle group CSA (or estimates; yellow; see Table 1) and $F_{max}$ (magenta). Continuous lines represent regression curves using a logarithmic relationship: $y = a \times \ln(x + 1)$. For muscle group CSA: $y = -3.57 \times \ln(x + 1)$ ($R^2$: 0.49). For $F_{max}$: $y = -6.96 \times \ln(x + 1)$ ($R^2$: 0.42). For comparison with the existing literature, dashed lines represent regression curves for the decline in muscle group CSA and $F_{max}$ reported in the comprehensive review of Marusic et al. (2021) (see main text for details). *B* and *D*, percentage changes (relative to baseline) of $F_{max}$ (magenta) and RFD (cyan), and their difference (grey). Percentage changes were also calculated on values relative to muscle group CSA or estimates (*C* and *E*). In 10 instances out of 12, the decline in RFD was greater than the decline in $F_{max}$. The longer the duration

of the muscle unloading protocol, the smaller the difference between the declines of $F_{max}$ and RFD. Relationships were modelled with the exponential (decaying) function of the form $y = a \times e^{-b\,x}$, constraining $a$ and $b > 0$, yielding for absolute values (D) the equation $y = 24.7 \times e^{-0.032\,x}$ ($R^2$: 0.30), and for normalised values (E) the equation $y = 27.8 \times e^{-0.023\,x}$ ($R^2$: 0.31). See main text for details on the fitting procedures, and of the confidence intervals of the outcome variables.

Fig. 3C reports the decline in the same measures normalised to muscle group CSA (or estimates; data points from individual studies reported in Table 3). Measures of the decline in RFD are more scattered than those of $F_{max}$, most likely due to the greater intrinsic variability of explosive strength indices (Buckthorpe et al., 2012). The differences between the percentage change of $F_{max}$ and RFD from the individual studies are reported in Fig. 3D (for absolute measures) and E (for measures relative to muscle group CSA). In 10 out of 12 instances, the decline in RFD was more pronounced than the decline in $F_{max}$. The longer the duration of the muscle unloading protocol, the smaller the difference between the decline in isometric maximal and explosive strength. Relationships were modelled with the exponential (decaying) function of the form $y = a \times e^{-b\,x}$, constraining $a$ and $b > 0$, yielding for absolute values (Fig. 3D) the equation $y = 24.7 \times e^{-0.032\,x}$ ($R^2$: 0.30; 95% confidence bounds for $a$ and $b$: [8.8, 40.0], [0.017, 0.048]), and for normalised values (Fig. 3E) the equation $y = 27.8 \times e^{-0.023\,x}$ ($R^2$: 0.31; 95% confidence bounds for $a$ and $b$: [12.0, 43.6], [0.007 0.039]). Time constants were 31.2 and 43.5 days, respectively.

Individual results divided by type of unloading (bed rest *vs.* immobilisation *vs.* unilateral lower limb suspension) are plotted in Fig. A3 in the Supporting information.

### Underlying neuromuscular mechanisms for the selective decline in explosive strength with disuse

With muscle mechanical unloading, the decline in $F_{max}$ is about twice that of muscle atrophy (cf. Fig. 3A), as evidenced by the negative constant multiplier of the logarithmic regressions (see also the Figs 2 and 3 in Marusic et al., 2021). That is, muscle disuse results in a decline in intrinsic muscle strength, which is well known and summarised in previous work (e.g. Marusic et al., 2021). With regard to explosive strength, impairments are greater than those observed for maximal strength (Fig. 3D and E). This represents a decline in the capability of the neuromuscular system to develop force fast, ascribed to mechanisms that affect RFD independently of $F_{max}$. The relative influence of such mechanisms on explosive strength decays with time, as in long-term muscle unloading (>40 days) the decline in explosive and maximal strength are almost similar (Fig. 3D and E). The mechanisms most likely fall within the determinants of explosive strength: the recruitment and firing rate of MUs, the likelihood of doublet discharges, muscle contra-

ctile properties, as well as musculo-tendinous stiffness and muscle belly gearing (see section 'Determinants of explosive strength'). As recently highlighted, such determinants are not discrete but lie within a neuro-mechanical continuum for strength production (Del Vecchio, 2023).

Despite the utmost importance for force control, only a handful of studies have focused on changes in MU recruitment and firing rate in response to muscle mechanical unloading in humans (Duchateau & Hainaut, 1990; Inns et al., 2022; Sarto et al., 2022; Seki et al., 2001b, 2007; Valli, Sarto, et al., 2024). Given that muscle unloading in many ways can be considered as the opposite stimulus of strength training (e.g. Duchateau & Enoka, 2002; Sale et al., 1982), it is not surprising that typical adaptations at the MU level with muscle unloading follow an opposite trend compared to those that have been reported after strength training (Table 4). Such a trend might, however, lack reliability if low- and high-threshold MUs are separately considered after muscle unloading (Valli, Sarto, et al., 2024).

In the only two studies that measured MUs relative recruitment threshold (defined as the %$F_{max}$ at which MUs start firing; Duchateau & Hainaut, 1990; Valli, Sarto, et al., 2024), this measure was higher after muscle unloading. Such an increase seems to depend on MU properties, as has been demonstrated for low-threshold MUs (i.e. recruited at less than 25% $F_{max}$; Valli, Sarto, et al., 2024; see also Fig. 5 of Duchateau & Hainaut, 1990), but not for high-threshold MUs (Valli, Sarto, et al., 2024). Interestingly, the relative recruitment threshold was found unchanged for MUs that were longitudinally tracked (Valli, Sarto, et al., 2024), potentially indicating that when the same MUs are investigated before and after the muscle unloading intervention, alterations in the relative recruitment threshold may not be significant. On the contrary, when considered in absolute terms (in N), MU recruitment threshold was decreased across all identified (and longitudinally tracked) MUs (Valli, Sarto, et al., 2024), as the contractile twitch force for all MUs is lower, and MUs are recruited earlier to attain the same level of force.

After muscle unloading, the relationship between force (as %$F_{max}$) and mean MU firing rate is less steep (see Fig. 5A of Seki et al., 2001b): MU firing rate is lower in MVCs (by 20%–40%) as well as when submaximal absolute or relative forces are targeted (Duchateau & Hainaut, 1990; Inns et al., 2022; Sarto et al., 2022; Seki et al., 2001a, b, 2007). A recent (high-density

**Table 4. Summary of study results examining MU adaptations after muscle unloading versus those after strength training (loading)**

|  | Muscle unloading* | Strength training** |
|---|---|---|
| Muscles examined | Adductor pollicis<br>First dorsal interosseus<br>Vastus lateralis | Abductor digiti minimi<br>Tibialis anterior<br>Vastus medialis / lateralis |
| Absolute recruitment threshold (N) | −[1] | −[7] |
| Relative recruitment threshold (%$F_{max}$) | +[1,2] | −[7] |
| Absolute derecruitment threshold (N) | −[1] | +[7] |
| Relative derecruitment threshold (%$F_{max}$) | =[1] | =[7] |
| Discharge rate at recruitment (Hz) | = | =[7] |
| Discharge rate at plateau (Hz) | −[1–6] | +[7–9] |
| Discharge rate at derecruitment (Hz) | −[1] | =[7] |
| Discharge rate modulation (Hz) | −[1] | +[7] |

$F_{max}$: isometric maximal voluntary contraction force or torque; MU: motor unit.
[1] Valli, Sarto, et al. (2024). Isometric ramp contractions of 5% $F_{max}$ s$^{-1}$ until 10, 25, and 50% $F_{max}$.
[2] Duchateau & Hainaut (1990). Isometric ramp contractions of 5% $F_{max}$ s$^{-1}$ until maximal force.
[3] Seki et al. (2001b). Isometric contractions at 20, 40, 60 and 80% $F_{max}$.
[4] Seki et al. (2007). Isometric maximal contractions.
[5] Inns et al. (2022).
[6] Sarto et al. (2022). Isometric contractions at 10 and 25% $F_{max}$.
[7] Del Vecchio, Casolo, et al. (2019).
[8] Casolo et al. (2020). Isometric ramp contractions of 5% $F_{max}$ s$^{-1}$ until 35, 50, and 70% $F_{max}$.
[9] Vila-Chã et al. (2010). Isometric contractions at 10 and 30% $F_{max}$.
*Results may differ if low- and high-threshold MUs are separately considered (Valli, Sarto, et al., 2024).
**Results refer to submaximal contractions and may be inconsistent for maximal efforts (Del Vecchio et al., 2024).

surface EMG) study, however, pointed out that this conclusion may not be generalisable to all MUs: after 10 days of unilateral lower limb suspension, early recruited (low-threshold) MUs displayed lower discharge rate during steady state submaximal contractions, while later recruited MUs (high-threshold) showed higher discharge rates (Valli, Sarto, et al., 2024). As for the relative recruitment threshold, when the same MUs are longitudinally tracked before and after the unloading intervention, their discharge rate may be unchanged regardless of MU type (Valli, Sarto, et al., 2024).

Given the results on MU recruitment threshold and mean firing rate, muscle unloading leads to a narrower firing rate modulation (Duchateau & Hainaut, 1990; Seki et al., 2001b). Such property, however, was recently found to be unchanged in higher-threshold MUs, and in MUs that were longitudinally tracked before and after disuse (Valli, Sarto, et al., 2024).

Overall, there is agreement across studies, at least for low-threshold MUs, that the relative recruitment thresholds are increased after muscle unloading, while mean firing rates and their modulation range are decreased. These findings seem to hold irrespective of the muscles studied (Table 4; adductor pollicis in Duchateau & Hainaut, 1990; first dorsal interosseus in Duchateau & Hainaut, 1990; Seki et al., 2001a, b, 2007; vastus lateralis in Inns et al., 2022; Sarto et al., 2022, and Valli, Sarto,

et al., 2024). The limitation to low-threshold MUs is consistent with animal studies, where twitch and tetanic maximal forces after immobilisation are mostly impaired in low-threshold fatigue resistant MUs (Cormery et al., 2005; Mayer et al., 1981; Petit & Gioux, 1993), and consistent with the interpretation that motoneurons and MUs experiencing the largest change in their normal activity patterns change the most in their properties (Cormery et al., 2005). One note of caution is necessary: when the same MUs are tracked (by 2-D cross-correlation with high-density EMG), changes in relative recruitment threshold, the mean firing rate, and the discharge rate modulation may be not significantly evident (Valli, Sarto, et al., 2024).

Modifications in the behaviour of MU properties after muscle unloading may be driven by changes in MU number and size, or alterations in the neuromuscular junction. Previous studies have found that MU numbers and sizes are unchanged with muscle unloading (Attias et al., 2020). In addition, the high transmission efficiency of the neuromuscular junction is preserved (Inns et al., 2022; Monti et al., 2021; Sarto et al., 2022), despite modifications in its integrity (e.g. increased molecular instability and signs of partial denervation; Monti et al., 2021; Sarto et al., 2022; see Sirago et al., 2023 for review) as early as 3 days after the onset of disuse (Demangel et al., 2017). This indicates that the net excitation of

the motoneuron pool for the same relative force ($\%F_{max}$) decreases with muscle unloading, which is corroborated by the widespread finding that the capacity to activate muscles during MVCs (measured with supramaximal peripheral nerve stimulation) is impaired after disuse (e.g. Campbell et al., 2019; Duchateau, 1995; Gondin et al., 2004; Semmler et al., 2000), even if contrasting findings have been reported (e.g. Clark, Manini et al., 2006; Seo et al., 2024). The remaining candidates for the observed changes in MU behaviour with muscle unloading are therefore altered net excitatory input to the motoneuron pool and/or motoneuron adaptations (neurophysiological properties and intrinsic excitability).

As it is for strength training (Del Vecchio et al., 2024; Škarabot et al., 2021), it is hard to pinpoint conclusively the specific sites of adaptations after muscle unloading that may yield lower net excitatory input to motoneurons. Previous research has shown inconsistent findings regarding changes in corticospinal excitability with muscle disuse, with reports of increased (e.g. Roberts et al., 2007), unchanged (e.g. Harmon et al., 2024; Seo et al., 2024), or decreased (e.g. Gaffney et al., 2021; Roberts et al., 2010) measures. Altered functional brain connectivity (Clouette et al., 2024; Demertzi et al., 2016), decreased cortical representation in the primary motor cortex of the unused muscles and decreased cortical thickness (grey matter) are common findings (Langer et al., 2012; Liepert et al., 1995), also reported in murine models of muscle unloading (Langlet et al., 2012; Mysoet et al., 2017; Viaro et al., 2014).

Another candidate for the reduced net excitatory input are the subcortical projections from the brainstem of the reticulospinal pathway. Several studies have found that increased input from the reticulospinal pathway acutely increases maximal (Anzak et al., 2011) and explosive (Škarabot et al., 2022) strength and adapts specifically to resistance training (Akalu et al., 2023; Atkinson et al., 2022; Glover & Baker, 2020). While the reticulospinal pathway has an important role in the recovery of motor function in clinical conditions such as spinal cord injuries (see Akalu et al., 2023 for review), and may be involved in age-related muscle weakness (Maitland & Baker, 2021), to the authors' knowledge there are no reports of reticulospinal tract plasticity with muscle unloading.

Neurophysiological properties of motoneurons themselves are sensitive to chronic changes in neuromuscular activity and inactivity (Dai et al., 2024; Gardiner et al., 2006). Murine models have shown that muscle unloading resulted in elevated motoneuron rheobase current, more depolarised spike threshold, faster time constants, lower cell capacitance, reduced afterhyperpolarisation amplitude, increased minimum current for steady state firing, and a rightward shifting of the frequency-current relationship (i.e. more current required to obtain a given repetitive firing frequency; Cormery et al., 2005; Dai et al., 2024). These findings

indicate that motoneurons became less excitable after muscle unloading.

One potential mechanism that modifies motoneuronal excitability is the neuromodulatory input of the descending monoaminergic system from the brainstem (Heckman et al., 2009). Such input generates persistent inward currents which amplify the responsiveness to the excitatory drive. The strength of these persistent inward currents, which can be estimated *in vivo* in humans from MU recruitment-derecruitment firing hysteresis, was reduced after 10 days of unilateral lower limb suspension (Martino et al., 2024), indicating that the lower neuromodulatory input to motoneurons after a period of muscle unloading may decrease their responsiveness to net excitatory inputs.

Overall, alterations in cortical areas and functional connections, modifications in the neural output from the corticospinal and other pathways (e.g. reticulospinal), lower motoneuron pool excitability from modified neurophysiological properties, together with lower neuromodulatory input, may be responsible for the increased relative recruitment thresholds and decreased mean firing rates of MUs after a period of muscle unloading. To better understand the locus and nature of these changes within the nervous system, further studies using neurophysiological techniques such as transcranial magnetic stimulation of the motor cortex (Bruce et al., 2023; Todd et al., 2003), and stimulation of the corticospinal tract (Martin et al., 2008; Škarabot et al., 2019; Taylor, 2006) could be conducted. For example, cortical voluntary activation (although applicable at present mostly on elbow flexors; e.g. Bruce et al., 2023; Ruggiero & McNeil, 2019) could be measured before and after muscle unloading protocols to determine changes in the capacity of the motor cortex to maximally activate the muscle. In addition, corticospinal tract stimulation could be used individually (McNeil et al., 2013), or paired with TMS (e.g. McNeil et al., 2011; Ruggiero, Yacyshyn, et al., 2018) to determine the influence of a period of muscle unloading on motoneuron pool excitability with and without the confound of unknown descending corticospinal drive. Such techniques could also be paired with startling auditory stimuli (see Atkinson et al., 2022 for review) to infer changes within the reticulospinal pathway after muscle disuse.

Based on these neural modifications reported after muscle disuse, it is very plausible that explosive strength is more impacted than maximal strength (Fig. 3). The neuromuscular system specifically modulates neural properties and corticospinal control to achieve the high RFD required in ballistic contractions. The first and most important property that is modified according to the RFD required is the recruitment threshold of MUs, i.e. the higher the RFD, the lower the MU recruitment threshold (Del Vecchio et al., 2018; Desmedt & Godaux, 1977a; Van

Cutsem et al., 1998). The recruitment characteristics of MUs and the firing rates of the recruited MUs determine the force-time relationship of a ballistic contraction. It is therefore intuitive that increased relative recruitment thresholds of MUs after muscle unloading, while retaining the net excitatory input to the motoneuron pool, lead to a slower recruitment rate of MUs at the beginning of the explosive effort, which in consequence affects explosive strength more than maximal strength. This hypothesis, however, has not been so far experimentally verified. The neural modulation within the corticospinal pathway before force onset is also dependent on the RFD of the upcoming contraction. Before ballistic contractions, when compared to slower ramp contractions, corticospinal excitability increases and corticospinal inhibition decreases closer and steeper relative to force onset, with no changes in spinal excitability (Baudry & Duchateau, 2021). These findings suggest that the motor cortex has a primary role in dictating RFD of the upcoming contraction, ensuring a synchronised descending excitatory input to achieve the rapid recruitment of MUs and high firing rates typical of explosive contractions (Baudry & Duchateau, 2021; Del Vecchio, Casolo, et al., 2019). Considering this, altered functional brain connectivity, decreased cortical representation of unused muscles, and the impaired capacity to activate muscles after unloading may affect explosive strength more than maximal strength. Finally, within the neuromuscular continuum, the changes in MU contractile kinetics typically experienced with muscle disuse may intrinsically limit RFD more than $F_{max}$. Beyond neural changes, muscle unloading is characterised by changes in the contractile and tendinous apparatuses. Within the contractile apparatus, several changes have been found with regard to excitation-contraction coupling (Fitts et al., 1991; Westerblad et al., 2000). For example, muscle unloading did not affect rate of fibre tension development (Fitts et al., 2007; Monti et al., 2021) but resulted in impaired intracellular $Ca^{2+}$ handling: reduced total amount of $Ca^{2+}$ in the sarcoplasmic reticulum, reduced responsiveness of its release channels, and a lower $Ca^{2+}$ release following depolarisation (Lamboley et al., 2016; Monti et al., 2021). It should, however, be acknowledged that findings regarding the shift and steepness of the force-$Ca^{2+}$ relationship are not uniform (Fitts et al., 2007; Hvid et al., 2011, 2013; Monti et al., 2021; Mounier et al., 2009; Widrick et al., 1998; Yamashita-Goto et al., 2001). Such inconsistencies in study outcomes can be explained by the dependency of results on the muscle and fibre type examined and the duration of the unloading protocol (Hvid et al., 2013; Monti et al., 2021; Yamashita-Goto et al., 2001). Within the tendon apparatus, lower stiffness and Young's Modulus are typically found after a period of muscle unloading (Roffino et al., 2021), e.g. −18% already after 14 days of lower limb immobilisation (Couppè

et al., 2012), with longer intervention durations typically corresponding to a greater decrease, e.g. −29% after 23 days of unilateral lower limb suspension (De Boer et al., 2007), −33% and −57% after 20 and 90 days of bed rest, respectively (Kubo et al., 2000, 2004; Reeves et al., 2005). It is plausible that this ensemble of changes in the muscular and tendinous apparatuses may lead to slower twitch contraction kinetics. Indeed, the muscle response to single, double, or triple supramaximal peripheral stimulation after a period of muscle mechanical unloading showed considerably lengthened contraction and half-relaxation times (Campbell et al., 2013; Clark, Fernhall et al., 2006; Cook et al., 2014; Davies et al., 1987; Seki et al., 2001a; Seynnes et al., 2010; Suetta et al., 2009; White et al., 1984). In such cases, a lowered MU discharge rate at all contraction levels can ensure greater twitch summation, minimizing the loss of force that results from muscle unloading (Seki et al., 2001a, b), at the expense of a more pronounced decline in explosive strength. To analytically test this hypothesis, we conducted simulations (MATLAB, MathWorks, Natick, MA, USA) with a Hill-type force model (Ding et al., 2002; Ruggiero et al., 2021; Wexler et al., 1997) used to reproduce the force-frequency relationship of animal and human muscles (see Supporting information, item A4 for details of the model). The parameters of the simulation were chosen to yield the same peak torque of single twitches and tetani at 10 and 100 Hz of the ankle dorsiflexors (Bruce et al., 2021; Ruggiero et al., 2019), represented by the black torque-time traces in Fig. 4*A*. The values of two parameters (a time constant and a scaling factor) were modified, reproducing slowed twitch contraction kinetics (longer contraction and half-relaxation time by 20% and 40%, respectively) and lower peak torque (by ∼18%), represented by the superimposed magenta torque-time traces in Fig. 4*A*. As depicted in Fig. 4*B*, except for very low stimulation frequencies (<4 Hz), with slowed twitch kinetics and lower peak torque, the same peak torque of the contraction can be attained at slightly lower frequencies of stimulation, or in other words, as represented by the percentage difference in peak torque between the two cases in Fig. 4*D*, at the same stimulation frequency peak torque is greater. Such peak torque is, however, achieved over a longer time, due to the much lower peak rate of torque development with slower twitch kinetics (Fig. 4*C* and *E*). Thus, the lower MU firing rates after muscle disuse most likely minimise the loss of force production when the muscle is activated at relatively low intensities, such as the force levels of muscles that are usually necessary for daily life activities (Kulmala et al., 2016, 2020), at the expense of a disproportionately lower RFD in ballistic contractions.

One important property that depends on the required RFD in functional movements is the neuromechanical delay. Previous research has highlighted that neuro-

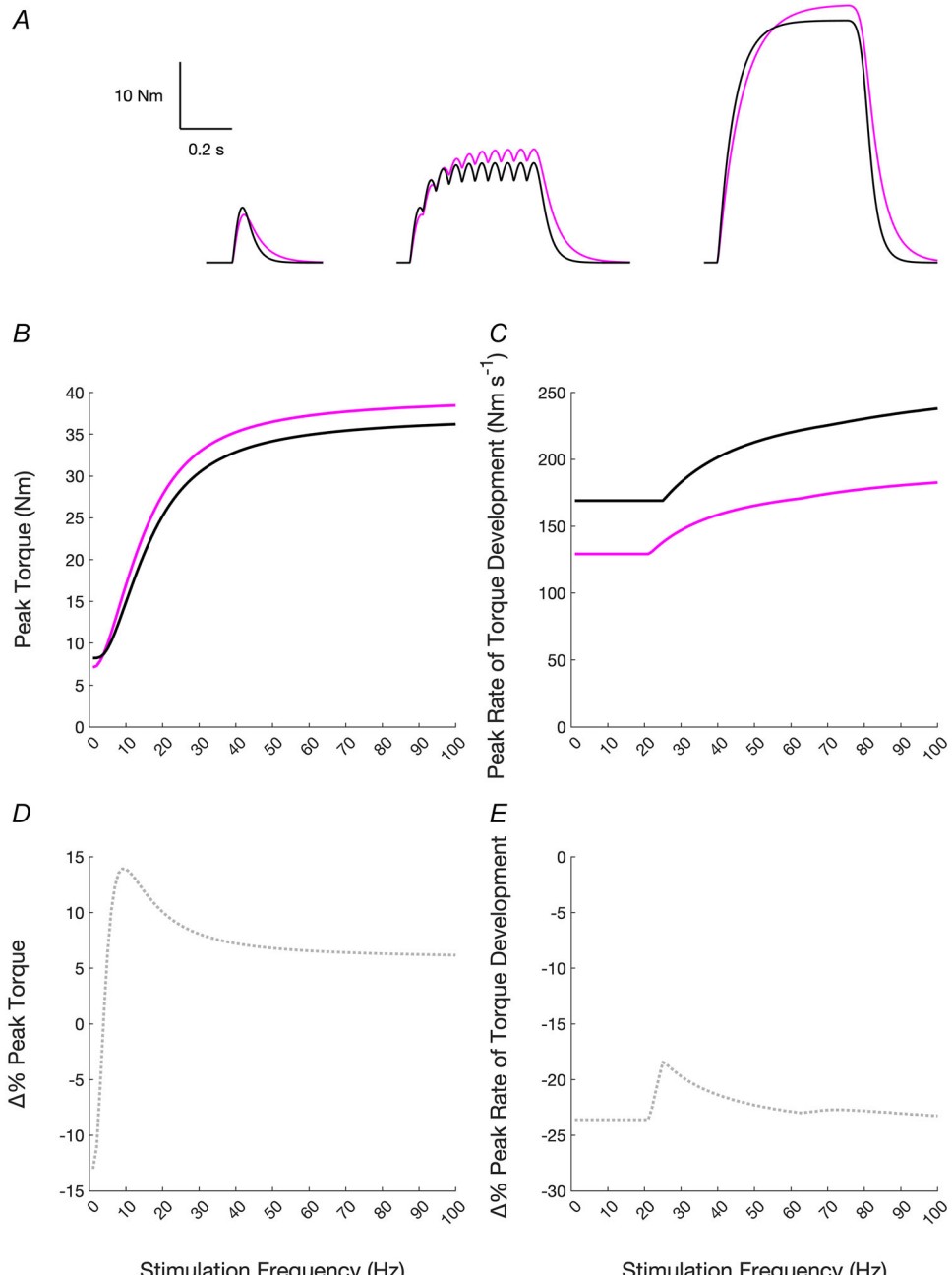

**Figure 4. Effect of slowed muscle contraction kinetics on tetanic peak torque and rate of torque development**

Muscle responses were modelled with a Hill-type force model (see main text and Supporting information, item A4 for details). *A*, torque time traces for a single twitch and tetani at 10 and 100 Hz (black lines), and resulting from the summation of twitches with slower contraction kinetics (longer contraction and half-relaxation time by 20% and 40%, respectively) and lower peak torque (by ∼18%; magenta lines). *B* and *D*, the resulting torque-frequency relationships, and peak torque percentage difference between the two conditions at different frequencies of stimulation (dotted grey line). Except for very low stimulation frequencies (<4 Hz), with slowed twitch kinetics, the same peak torque could be attained at slightly lower frequencies of stimulation, or in other words, as represented in *D*, at the same stimulation frequency, peak torque is greater. *C* and *E*, the resulting peak rate of torque development at the different frequencies, and the percentage difference of peak rate of torque development between the two simulations (dotted grey line). With slowed twitch contraction kinetics, peak rate of torque development is much lower at all frequencies, despite higher peak torque.

mechanical delay (i.e. the latency between neural drive to the muscle and force during voluntary contractions with variable force requirements) decreases non-linearly in the form of a negative exponential function with RFD: the higher the RFD, the lower the neuromechanical delay (Fig. 3*B* of Del Vecchio et al., 2018; Fig. 2 of Ùbeda et al., 2017). A lower neuromechanical delay facilitates accuracy of force at faster contraction rates (Del Vecchio et al., 2018). The ensemble of changes from muscle unloading within the neuromechanical continuum (increased relative recruitment threshold and decreased firing rate of MUs, slower rate of twitch tension development, and lower tendon stiffness) may preferentially lengthen the neuromechanical delay when high RFD for fast accuracy of force is needed. This calls for further caution when daily life activities are resumed after a period of muscle unloading, as a slower capacity to accurately react to perturbations is intrinsically present.

### Additional considerations

The following factors should be considered for a thorough understanding of the main findings in the present review: (1) distinction between weight-bearing and not weight-bearing muscles and flexor and extensor muscles, and (2) methodological considerations regarding explosive contractions and determination of RFD between studies.

Most of the literature included in the present review concerns weight-bearing muscle groups (i.e. knee extensors and ankle plantarflexors), with only one study looking at non-weight-bearing muscles (i.e. elbow flexors, Valdes et al., 2020). It is a common finding that muscle unloading has a greater impact on atrophy and strength loss in weight-bearing than non-weight-bearing muscles (Bass et al., 2021; Campbell et al., 2019; Marusic et al., 2021). Due to the low number of studies on non-weight-bearing muscles it remains to be determined whether the relationship found between declines in explosive and maximal strength is similar or different between the two muscle group types. In addition, given the limited amount of data available, no distinction can be made between flexor and extensor muscles for which it is known that motoneuronal characteristics may differ (e.g. Yachyshyn et al., 2018). Whether such differences affect explosive strength independently of maximal force production needs to be determined.

The second consideration concerns the methodology used to determine explosive strength. From the studies included in the present review (Table 1), different methods to determine contraction force onset were used (e.g. 3% $F_{max}$ in Hvid et al., 2013; 2 Nm above baseline in De Boer et al., 2007; and other authors have not specified a criterion). Yet, the moment of force onset in explosive contractions may greatly affect RFD outcome variables, and thus standardised procedures for its selection should be used, with the gold-standard being manual detection (Tillin et al., 2013b). Moreover, different joint positions were used in different studies, which may affect muscle fascicle length and in turn the RFD relative to $F_{max}$ (Hager et al., 2020). In addition different methods were used to calculate RFD (e.g. RFD between 0 and 100 ms from onset by De Boer et al., 2007; the derivative between 10% and 60% $F_{max}$ by Kubo et al., 2000; the maximal first force derivative value by Mulder et al., 2009; the time to reach 63% $F_{max}$ by Monti et al., 2021 and Sarto et al., 2022). There is a need to establish and report consistent methodologies and variables. For example, if not in the main article file, authors could report in the Supporting information data for all participants, both as raw values and normalised to $F_{max}$, peak RFD, the RFD and impulse between 0 and 50, 0 and 100, and 0 and 150 ms, and the force-time function from force onset. Such consistent reporting could reduce inconsistencies in data analyses that add variability between studies to a measure like RFD that is already intrinsically variable (Buckthorpe et al., 2012).

### Future directions

Several issues remain to be addressed regarding the effect of muscle disuse on explosive and maximal strength. As already mentioned, future studies could use stimulation techniques to determine the effect of muscle unloading on cortical voluntary activation, motoneuron excitability in humans, or the reticulospinal tract. Given the recent availability and non-invasive nature of high-density surface EMG systems and of analysis methods (e.g. Valli, Ritsche, et al., 2024), the effect of muscle unloading on the recruitment and firing rate of MUs in explosive efforts should be experimentally verified. Furthermore, eccentric *vs*. concentric explosive contractions, which differ in the neural determinants of early RFD (Tillin et al., 2012b, 2018) should be studied, to determine whether the impairment of RFD with muscle unloading is specific to different explosive contraction modalities. Finally, as highlighted above (see 'Additional Considerations' section), future research should study explosive strength after muscle mechanical unloading in weight-bearing *vs*. non-weight-bearing muscles, and in flexor *vs*. extensor muscles, to determine whether the effect of muscle disuse on explosive strength and MU behaviour differ between different types of motoneuron pools.

Increasing our knowledge of the neuromuscular factors determining the faster decline in explosive relative to maximal strength with muscle disuse is critical to establish more effective exercise paradigms and countermeasures to minimise such detrimental effects. For example, ballistic strength training as well as sensorimotor training, which

are both known to selectively target explosive strength based on different neural mechanisms (e.g. Gruber et al., 2007), could serve as exercise countermeasures. A better understanding of how to tailor such exercise countermeasures to the needs of individuals will help to preserve or recover functional performance in athletes, patients, the elderly, and in other groups that experience periods of muscle unloading.

## Conclusions

In the present narrative review, we summarised findings from muscle mechanical unloading (bed rest, immobilisation, and unilateral lower limb suspension; from 4 to 90 days) on isometric maximal and explosive strength. The results showed that explosive strength decreases at a faster rate than maximal strength. The greater the duration of the muscle mechanical unloading protocol, the smaller the difference between the decline in the two measures. Higher recruitment threshold and lower firing rates of motor units, together with slowed twitch contraction kinetics, impaired excitation-contraction coupling and decreased tendon stiffness, may explain the greater decline in explosive relative to maximal strength with muscle disuse. A profound understanding of the impairments associated with muscle unloading and their functional outcomes is critical to develop exercise countermeasures that are efficient in mitigating functional declines and can serve as the building blocks of training interventions that are able to fully restore neuromuscular performance after a period of muscle disuse.

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

## Additional information

### Competing interests

There are no competing interests to declare.

### Author contributions

Both authors contributed to the conception of the present review. LR collected information from individual studies and drafted the manuscript, revised the manuscript critically for important intellectual content, and have read and approved the final submission. All persons designated as authors qualify for authorship, and all those who qualify for authorship are listed.

### Funding

L.R. is supported by the Alexander von Humboldt Foundation.

### Acknowledgements

### Keywords

bed rest, explosive strength, limb immobilisation, muscle disuse, rate of force development, unilateral lower limb suspension, unloading

### Supporting information

Additional supporting information can be found online in the Supporting Information section at the end of the HTML view of the article. Supporting information files available:

**Peer Review History**
**Supplementary material**

