## [Peer Review History · The Journal of Physiology]

Neuromuscular mechanisms for the fast decline in rate of force development with muscle disuse - a narrative review.

Luca Ruggiero and Markus Gruber

DOI: 10.1113/JP285667

Corresponding author(s): Luca Ruggiero (luca.ruggiero@uni-konstanz.de)

The following individual(s) involved in review of this submission have agreed to reveal their identity: Per Aagaard (Referee #2)

Review Timeline:

Submission Date:	21-Jun-2024
Editorial Decision:	30-Jul-2024
Revision Received:	11-Sep-2024
Accepted:	27-Sep-2024

Senior Editor: Laura Bennet

Reviewing Editor: Christoph Centner

Transaction Report:

Dear Dr Ruggiero,

Re: JP-TR-2024-285667 "Neuromuscular mechanisms for the fast decline of rate of force development with muscle disuse."
by Luca Ruggiero and Markus Gruber

Thank you for submitting your manuscript to The Journal of Physiology. It has been assessed by a Reviewing Editor and by 2 expert referees and we are pleased to tell you that it is acceptable for publication following satisfactory revision.

ABSTRACT FIGURES: Authors may use The Journal's premium BioRender account to create/redraw their Abstract Figures (and any other suitable schematic figure). Information on how to access this account is here: <https://physoc.onlinelibrary.wiley.com/journal/14697793/biorender-access>.

REVISION CHECKLIST: Upload a full Response to Referees file. To create your 'Response to Referees' copy all the reports, including any comments from the Senior and Reviewing Editors, into a Microsoft Word, or similar, file and respond to each point, using font or background colour to distinguish comments and responses and upload as the required file type.

We look forward to receiving your revised submission.

Yours sincerely,

Laura Bennet
Senior Editor

EDITOR COMMENTS

Reviewing Editor:

The authors provide a comprehensive narrative review focusing on the neurophysiological mechanisms behind the accelerated loss of rapid force characteristics in response to mechanical unloading. I encourage the authors on a more focused discussion highlighting both the „what we know" but also the „what we dont know" on neural mechanisms behind this phenomenon.

Further, I agree with reviewer 2, that the word „narrative" (review) should be incorporated in the abstract and text to make clear to the reader that this is no systematic review.

Please also see 'Required Items' below.

REFEREE COMMENTS

Referee #1:

There are many basic concepts derived from Newtonian mechanics. Numerous reports and books already summarize these concepts and include the equations.

I also strongly recommend condensing the introduction to highlight the most salient points.

From Figure 2, the RFD is constant, and it differs from power (W). I suggest adding better examples to show the dependence/relationship of RFD with power.

182. They are better described as physically (not mathematically) related. These are physical concepts derived from Newton's laws of motion.

Aagaard, among others, was the first to show this. Citing much later work is inappropriate. Also, Aagaard (2002) clearly described the concepts of momentum, torque, impulse, etc., which are mainly what has been described here, but with many probably unnecessary equations.

261-266. There is absolutely no evidence to show that there are different determinants depending on the time elapsed from force onset. Please see Fig. 3D in Del Vecchio 2023 (ESSR). It is all a continuum. The cited work shows different relations mainly because they use metrics that do not reflect the actual neural drive.

This is a significant problem because it dramatically changes how the review is then shaped. It also seems to convey the wrong message that neural adaptations are "early" and muscular adaptations are "late," in terms of disuse/training/etc. There is really no evidence about this but just weak studies showing data from the RMS of the EMG to say that neural adaptations are not important at later stages. Actually, there is a lot of research showing that mortality is highly related to maximal grip strength and that grip strength has a strong neural determinant.

I think the neural adaptations are highly overlooked in this review, especially in the second part where there is a lot of focus on biochemical and MTU stiffness.

This review seems to me to basically just summarize the line of thought of several researchers without providing anything new from a fundamental perspective. I really liked the equations (although I am not sure they are very relevant for this review and perhaps better suited to another separate publication), but there is a missing "spicy" discussion on what we do not know about changes in neuromuscular function.

I think the work can be accepted if the review during the next iteration is more focused and tries to make a new argument on this important topic. At the moment, the manuscript is very narrative and discusses findings that the community already knows. In any case, I want to congratulate the authors on summarizing a lot of relevant literature.

Referee #2:

This narrative review describes the underlying neural and musculotendinous mechanisms that are major candidates to explain the more pronounced and faster decline of isometric rapid muscle force (often termed explosive force, operationally defined as the rate of force development, RFD) compared to maximal strength in response to short-term muscle disuse/unloading. Factors suggested to explain the selective decline in explosive force (RFD) relative to maximal force with muscle mechanical unloading comprise elevated MU recruitment thresholds and higher MU firing rates along with impaired intracellular Ca²⁺ handling, and lower tendon stiffness.

The Authors provide an interesting and informative review on the effect of muscle unloading on the neuromechanics of rapid force production (RFD), which integrates experimental findings from various subfields of experimental physiology, to improve our understanding of the rapid decline in RFD observed with short-term disuse in humans. Based on these considerations, (more) effective countermeasures may be employed i.e. in old adults, clinical patients and athletes to whom losses in RFD may be particular detrimental for functional capacity.

However, a number of points need to be addressed, as elaborated below.

Specific Comments

(1) Abstract line 27.

Suggest to add 'narrative' to inform the Reader that this is not a systematic literature review:

"... this narrative review ..."

(2) Line 51

Suggest to explain/elaborate the term 'explosive strength' to the Reader, for instance like this:

"... produce the highest force in the minimum time, referred to as rapid force capacity (rate of force development: RFD) or explosive strength ..."

(3) Line 52

Suggest to also cite Rodríguez-Rosell et al. 2018, which is already listed in the References section.

Rodríguez-Rosell D, Pareja-Blanco F, Aagaard P, José González-Badillo JJ. Physiological and methodological aspects of rate of force development assessment in human skeletal muscle. *J. Clin. Physiol. Funct. Imaging* 38, 743-762, 2018

(4) Lines 56-57

Consider rephrasing to improve the English syntax, suggestion:

"... physiological reasons behind its plasticity with training (e.g., Aagaard et al., 2002; Gruber et al., 2004; Van Cutsem et al., 1998) or lack hereof (e.g., Del Vecchio et al., 2022) ..."

(5) Lines 74-75

Suggest to also cite Suetta et al. 2009 and Hvid et al. 2014 at this point in the text, especially since these studies are already listed in the List of References.

(6) Lines 82-83

It is stated that "... although this ability [RFD] is more relevant than maximal strength to human performance ..."

(i) suggest to replace "is" with "may be":

"... although this ability may be more relevant than maximal strength to human performance ..."

(ii) Suggest to include references at the end of this sentence to support the statement, i.e. Maffiuletti et al. 2010 (already in the list of References) as well as Orssatto et al. 2020 and Lomborg et al. 2022

Orssatto LBR, Bezerra ES, Schoenfeld BJ, Diefenthaler F. Lean, fast and strong: Determinants of functional performance in the elderly. *Clin Biomech (Bristol, Avon)* 78, 105073, 2020

Lomborg SD, Dalgas U, Hvid LG. The importance of neuromuscular rate of force development for physical function in aging and common neurodegenerative disorders - a systematic review. *J Musculoskelet Neuronal Interact* 22, 562-586, 2022

(7) Line 89

Suggest to add 'narrative', cf. above Comment 1:

"... this narrative review ..."

(8) Line 85

To improve the English syntax, suggest to replace 'work out' with 'address':

".... To address the specific mechanisms underlying the decline in explosive ..."

(9) Line 99

suggest to insert 'physiological':

"... the impact on its physiological determinants ..."

(10) Line 105

Suggest to inform the Reader that this equation quantifies the production of impulse

"... to the following relationship describing the production of impulse: ..."

(11) Lines 115-116

To facilitate the concept of impulse to the Reader, suggest to cite Rodríguez-Rosell et al. 2018 (their Fig.4):

"... to produce the greatest impulse, the highest amount of force in the minimum time must be developed (Rodríguez-Rosell et al. 2018, their Fig.4) ..."

(12) Line 140

To be more consistent, suggest to insert 'produced during':

"... the impulse produced during the first 250 ms ..."

(13) Line 159

suggest to add 'postural':

"... in elderly, indices of postural balance correlated with RFD ..."

Similar relationships also have been observed in young adults (i.e. Jakobsen et al. 2011), which might deserve mentioning.

Jakobsen MD, Sundstrup E, Krstrup P, Aagaard P. The effect of recreational soccer training and running on postural balance in untrained men. *Eur. J. Appl. Physiol.* 111, 521-530, 2011

(14) Lines 171-172

The use of RFD testing to guide a safe return to sport following musculoskeletal injury is an emerging yet important aspect in athlete rehabilitation, which is only rarely addressed in the literature. Consequently, consider also citing Jordan et al. 2023 to support/elaborate on this approach.

Jordan MJ, Aagaard P, Bishop C, McLean Z, Morris N, Boon-van Mossel N, Pasanen K, da Silva Torres R, Herzog W. Explosive strength and stretch-shortening-cycle capacity during ACL rehabilitation - Mechanical biomarkers for return to sport and performance readiness. *ASPETAR Sports Med J* 12, 324-331, 2023

(15) Line 223

I fail to see the relevance/information of this partial derivative of power relative to RFD, as it doesn't explain any features of the right-most graph (Power-time curve) shown in Figure 2.

I suggest the Authors to instead use the equation in line 221 to derive/develop the analytical relationship for the rate of power development, i.e. $dW[\dot{t}]/dt$. This relationship may be considered more interesting/relevant as it will describe the shape (slope) of the power-time curve shown in Figure 2, thus extending and explaining previous observations of training-induced improvements in the rate of power development reported in the literature (Jakobsen et al. 2012), which might also be relevant to discuss at this point of the text.

Jakobsen MD, Sundstrup E, Randers MB, Kjær M, Andersen LL, Krstrup P, Aagaard P. The effect of strength training,

(16) Line 86

Suggest to insert 'prolonged periods of'

"... even during prolonged periods of detraining ..."

(17) Lines 234-236

It is stated that

"... ii) in dynamic movements, RFD (from a muscle or a muscle group) is affected by muscle-length and contraction-velocity dependent mechanisms (Hahn et al., 2017; Lieber & Ward, 2011) ..."

In addition, RFD is also affected by acute/instantaneous changes in fascicle pennation angle (shifts in gearing ratio) (Van Hooren et al. 2024), which perhaps should be briefly mentioned at this point in the text.

Van Hooren B, Aagaard P, Monte A, Blazevich AJ. The role of pennation angle and architectural gearing to rate of force development in dynamic and isometric muscle contractions. Scand. J. Med. Sci. Sports 34, e14639, 2024

(18) Lines 254

suggest to elaborate / be more concise:

"... high MU firing rates ..."

(19) Line 156

Suggest to insert ' via increased rates of'

"... promotion for proteogenesis in myonuclei via increased rates of transcription and translation [21] ..."

(20) Line 259

'increase' seems to be missing:

"... MUs to maximise their initial firing rate, and in turn increase RFD (Aoki et al., 2002 ..."

(21) Line 274

Suggest to change "... did not influence ..." into "... may not influence ..." since MU synchronization may not be very clearly distinguished by current HDEMG analysis algorithms.

(22) Line 280

Should say "... Bojsen-Møller et al. ..."

(23) Line 281

Suggest to also cite seminal study by Lichtwark and coworkers 2016.

Mayfield DL, Cresswell AG, Lichtwark GA. Effects of series elastic compliance on muscle force summation and the rate of force rise. *J Exp Biol* 219, 3261-3270, 2016

(24) Lines 287-291

In terms of the influence of muscle architectural gearing on RFD, suggest to also cite recent paper by Van Hooren et al. 2024.

Van Hooren B, Aagaard P, Monte A, Blazevich AJ. The role of pennation angle and architectural gearing to rate of force development in dynamic and isometric muscle contractions. *Scand. J. Med. Sci. Sports* 34, e14639, 2024

(25) Lines 309-310

It is stated that "... Such discrepancy between eccentric and concentric contractions comes from neural inhibition of force production in the former modality ..."

To support this notion, suggest to cite reference(s) that have addressed the presence of neural inhibition during maximal eccentric muscle actions, i.e. Aagaard 2018.

Aagaard P. Spinal and supraspinal control of motor function during maximal eccentric muscle contraction: Effects of resistance training. *J Sport Health Sci* 7, 282-293, 2018

(26) Line 319

Suggest to exchange 'needed' with 'relevant':

"... However, when relevant, reference will be made to dynamic explosive movements ..."

(27) Lines 350-373

Suggest to delete paragraph listed in lines 350-373, to reduce the (somewhat excessive) length of the manuscript.

(28) Line 418

Suggest to rephrase to improve the English syntax

"... applying a logarithmic equation

(29) Line 454

Figure 3. For each data type, single data points represent a single study? Please make this more clear to the Reader.

(30) Line 475

Should say "... IN EXPLOSIVE STRENGTH WITH DISUSE ..." ?

(31) Line 476

Exchange 'of' with 'in':

"... in Fmax ..."

(32) Line 477

To assist the Reader, suggest to add '(cf. Figure 3A)':

"... atrophy (cf. Figure 3A), as evident ..."

(33) Line 506

To improve consistency, suggest to add 'caused by changes in'

" ... or caused by changes in intrinsic motoneuronal properties ..."

(34) Line 507

Suggest to insert 'synaptic':

"... equal synaptic input to motoneurons ..."

(35) Line 566

Suggest to add 'however'

"... production, however at the expense ..."

(36) Line 624

Suggest to add 'level':

"... same excitation level. Combined ..."

(37) Line 630

To be more concise, suggest to add 'reduced':

"... found to decline, e.g., 18% reduced already after 14 days ..."

(38) Lines 631-632

It is stated that "... with longer intervention durations typically corresponding to a greater decrease (De Boer et al., 2007; Kubo et al., 2000; Kubo et al., 2004; Reeves et al., 2005; Roffino et al., 2021) ..."

How much decrease (% decline) in how many days? Please include this information in the text.

(39) Line 633

Suggest to use the pluralis form:

"... such declines in tendon stiffness ..."

(40) Lines 636-677

Suggest to omit this entire section (lines 636-677), to reduce the (very long) length of the review.

(41) Line 692

At the end of this section (Future directions), the Authors should consider including 2-4 sentences addressing how an increased knowledge of the factors mentioned/discussed in the present review may be used to establish more effective countermeasures (exercise paradigms, NMES, passive/active BFR, etc) to minimize the detrimental effect of short-term disuse on RFD, to thereby prevent/attenuate impairments in the functional performance of athletes, elderly/old adults and clinical patients, respectively.

REQUIRED ITEMS

- Please include an Abstract Figure file, as well as the Figure Legend text within the main article file. The Abstract Figure is a piece of artwork designed to give readers an immediate understanding of the Review Article and should summarise the main conclusions. If possible, the image should be easily 'readable' from left to right or top to bottom. It should show the physiological relevance of the Review so readers can assess the importance and content of the article. Abstract Figures should not merely recapitulate other figures in the Review. Please try to keep the diagram as simple as possible and without superfluous information that may distract from the main conclusion of the Review. Abstract Figures must be provided by authors no later than the revised manuscript stage and should be uploaded as a separate file during online submission labelled as File Type 'Abstract Figure'. Please ensure that you include the figure legend in the main article file. All Abstract Figures will be sent to a professional illustrator for redrawing and you may be asked to approve the redrawn figure before your paper is accepted.

- Please upload separate high quality figure files via the submission form.

- Author profile(s) must be uploaded via the submission form. Authors should submit a short biography (no more than 100 words for one author or 150 words in total for two authors) and a portrait photograph of the two leading authors on the paper. These should be uploaded and clearly labelled together in a Word document with the revised version of the manuscript. Any standard image format for the photograph is acceptable, but the resolution should be at least 300 DPI and preferably more. A group photograph of all authors is also acceptable, providing the biography for the whole group does not exceed 150 words.

- Please ensure that the Article File you upload is a Word file.

END OF COMMENTS

Confidential Review

21-Jun-2024

Reviewing Editor:

The authors provide a comprehensive narrative review focusing on the neurophysiological mechanisms behind the accelerated loss of rapid force characteristics in response to mechanical unloading. I encourage the authors on a more focused discussion highlighting both the „what we know" but also the „what we dont know" on neural mechanisms behind this phenomenon.

Further, I agree with reviewer 2, that the word „narrative" (review) should be incorporated in the abstract and text to make clear to the reader that this is no systematic review.

Please also see 'Required Items' below.

We thank the Editor and the reviewers for the helpful and constructive comments on the manuscript. In the section “Underlying neuromuscular mechanisms for the selective decline in explosive strength with disuse” we have explicitly reported neural adaptations with muscle disuse that affect the net excitatory input to motoneurons and their intrinsic excitability (e.g., corticospinal and reticulospinal pathways, neuromodulatory input to motoneurons and changes in motoneurons neurophysiological properties), and which would explain a greater decline of explosive relative to maximal strength (the “what we know” part). In addition, we have explicitly reported what is still not known, and where gaps for further research exist.

The word “narrative” has been added in all instances where we refer to our work, as well as in the title.

Finally, the “Required Items” have been provided.

Referee #1:

There are many basic concepts derived from Newtonian mechanics. Numerous reports and books already summarize these concepts and include the equations.

Please see answer to the comment after next.

I also strongly recommend condensing the introduction to highlight the most salient points.

The introduction has been shortened by about half a page. We had to include further info and references suggested by the second reviewer. All together, now the introduction is 1 and half page long if references are not considered.

From Figure 2, the RFD is constant, and it differs from power (W). I suggest adding better examples to show the dependence/relationship of RFD with power.

The simulation with a linear actuator and constant RFD gives a simple opportunity for the reader to understand that RFD, power, and rate of power development are tightly related. Given that the simulation was found of value from Reviewer 2, who suggested to add a further equation to represent the coupling between RFD and rate of power development, we opted to keep the simulation with the linear actuator in. We have however explicitly listed in the end of this section that “real” contractions have a RFD function that is not constant (lines 251-253 of the revised manuscript).

182. They are better described as physically (not mathematically) related. These are physical concepts derived from Newton's laws of motion.

We have modified all instances where we wrote “mathematically” with “physically” (lines 203, 217 and 261 of the revised manuscript).

Aagaard, among others, was the first to show this. Citing much later work is inappropriate. Also, Aagaard (2002) clearly described the concepts of momentum, torque, impulse, etc., which are mainly what has been described here, but with many probably unnecessary equations.

We thank the reviewer for pointing this out. We had not included the reference to the important work of Aagaard et al. (2002) in this section. The work is now cited.

In his seminal work, Aagaard et al. (2002) reported that impulse is worth measuring from the force trace of isometric contractions because it equals the change in linear and angular momentum of the lower leg, and therefore the change in linear and angular velocity if the leg was allowed to move. The work of Minetti (2002) and Zamparo et al. (2002), conducted simulations with a linear actuator to show how changes in muscle maximal force or cross-sectional affect power of the movement. Their work however was not focused on RFD. We conducted a similar simulation solving equations explicitly for RFD, to show that changes in RFD are tightly coupled to power and rate of power development.

The much later work cited in this section regards articles that have measured lower limb extension power after muscle disuse. While power is not a direct measure of fast force production, our simulation shows that when the movement starts from stationary conditions, and it is not constrained (e.g., squat jumps or explosive leg press), maximal power and rate of power development are physically related to RFD. Therefore, even if such studies did not regard explosive force, they regarded muscle disuse and something that can be considered close to explosive force (power), and in our opinion it was fair to cite their work and to acknowledge it.

261-266. There is absolutely no evidence to show that there are different determinants depending on the time elapsed from force onset. Please see Fig. 3D in Del Vecchio 2023 (ESSR). It is all a continuum. The cited work shows different relations mainly because they use metrics that do not reflect the actual neural drive.

We agree with the reviewer for this and the following points. The wording was indeed misleading, and it has been changed (lines 280-286 of the revised manuscript). We have modified the manuscript to make sure that the message that comes across is that RFD is a continuum and not determined by discrete mechanisms (see also answers to the comments below).

This is a significant problem because it dramatically changes how the review is then shaped. It also seems to convey the wrong message that neural adaptations are "early" and muscular adaptations are "late," in terms of disuse/training/etc. There is really no evidence about this but just weak studies showing data from the RMS of the EMG to say that neural adaptations are not important at later stages. Actually, there is a lot of research showing that mortality is highly related to maximal grip strength and that grip strength has a strong neural determinant.

I think the neural adaptations are highly overlooked in this review, especially in the second part where there is a lot of focus on biochemical and MTU stiffness.

The data collected from previous studies highlight that there is greater difference in the decline of explosive vs. maximal force early with muscle unloading. However, our intention is not to divide early vs. late adaptations. We have now made sure that there are no misunderstandings in the text regarding this issue.

We have extensively modified the section "Underlying neuromuscular mechanisms for the selective decline in explosive strength with disuse" to focus on neural adaptations underpinning changes in MUs behaviour (lines

477-705). The muscular and tendinous adaptations have been framed in the neuromechanical continuum, and are not considered anymore as separate entities. The neural adaptations have been distinguished in those that affect net excitatory input to motoneurons (e.g., corticospinal and reticulospinal tract) and those that modify motoneurons intrinsic excitability (neuromodulatory input or changes in the neurophysiological properties of motoneurons). We have also explicitly reported what is still not known, and where gaps for further research exist.

This review seems to me to basically just summarize the line of thought of several researchers without providing anything new from a fundamental perspective. I really liked the equations (although I am not sure they are very relevant for this review and perhaps better suited to another separate publication), but there is a missing "spicy" discussion on what we do not know about changes in neuromuscular function.

As underlined in the comment above, we have extensively modified the section "Underlying neuromuscular mechanisms for the selective decline in explosive strength with disuse" providing a discussion on the neural mechanisms with muscle disuse underpinning declines in explosive strength, and on what is not known yet.

I think the work can be accepted if the review during the next iteration is more focused and tries to make a new argument on this important topic. At the moment, the manuscript is very narrative and discusses findings that the community already knows. In any case, I want to congratulate the authors on summarizing a lot of relevant literature.

We thank the author for the feedback, that we think considerably improved the manuscript.

Referee #2:

This narrative review describes the underlying neural and musculotendinous mechanisms that are major candidates to explain the more pronounced and faster decline of isometric rapid muscle force (often termed explosive force, operationally defined as the rate of force development, RFD) compared to maximal strength in response to short-term muscle disuse/unloading. Factors suggested to explain the selective decline in explosive force (RFD) relative to maximal force with muscle mechanical unloading comprise elevated MU recruitment thresholds and higher MU firing rates along with impaired intracellular Ca²⁺ handling, and

lower tendon stiffness.

The Authors provide an interesting and informative review on the effect of muscle unloading on the neuromechanics of rapid force production (RFD), which integrates experimental findings from various subfields of experimental physiology, to improve our understanding of the rapid decline in RFD observed with short-term disuse in humans. Based on these considerations, (more) effective countermeasures may be employed i.e. in old adults, clinical patients and athletes to whom losses in RFD may be particular detrimental for functional capacity.

However, a number of points need to be addressed, as elaborated below.

We thank the reviewer for the useful and constructive comments. Answers are reported below in red under each respective comment.

Specific Comments

(1) Abstract line 27.

Suggest to add 'narrative' to inform the Reader that this is not a systematic literature review:

"... this narrative review ..."

The word "narrative" review has been added in the abstract as suggested, in the title, and in other instances in the text. Given the extensive comments of reviewer #1, the abstract has been modified.

(2) Line 51

Suggest to explain/elaborate the term 'explosive strength' to the Reader, for instance like this:

"... produce the highest force in the minimum time, referred to as rapid force capacity (rate of force development: RFD) or explosive strength ..."

As suggested, we have added "rapid force capacity" for the reader to better elaborate on the concept of explosive strength (line 75 of revised manuscript). However, we refer to rate of force development (RFD) later in the review (line 165 of the revised manuscript) as RFD is a metric that is used to quantify rapid force capacity or explosive strength, and we wanted to distinguish the physical capacity from the metric that is commonly used to quantify it.

(3) Line 52

Suggest to also cite Rodríguez-Rosell et al. 2018, which is already listed in the References section.

Rodríguez-Rosell D, Pareja-Blanco F, Aagaard P, José González-Badillo JJ. Physiological and methodological aspects of rate of force development assessment in human skeletal muscle. J. Clin. Physiol. Funct. Imaging 38, 743-762, 2018

The citation has been added (line 75 of the revised manuscript).

(4) Lines 56-57

Consider rephrasing to improve the English syntax, suggestion:

"... physiological reasons behind its plasticity with training (e.g., Aagaard et al., 2002; Gruber et al., 2004; Van Cutsem et al., 1998) or lack hereof (e.g., Del Vecchio et al., 2022) ..."

We thank the reviewer for the suggestion, the sentence has been modified (lines 79-81 of revised manuscript).

(5) Lines 74-75

Suggest to also cite Suetta et al. 2009 and Hvid et al. 2014 at this point in the text, especially since these studies are already listed in the List of References.

We agree with the reviewer that those are seminal studies regarding the effect of muscle disuse on neuromuscular function. In lines 74-75 however (lines 97-98 of the revised manuscript) only reviews have been purposely cited, reason why we have not included reference to the work of Suetta et al. (2009) and Hvid et al. (2014), as well as other work. If however, the reviewer feels that the addition of these two references is of primary importance here, we will cite them.

(6) Lines 82-83

It is stated that "... although this ability [RFD] is more relevant than maximal strength to human performance ..."

(i) suggest to replace "is" with "may be":

"... although this ability may be more relevant than maximal strength to human performance ..."

(ii) Suggest to include references at the end of this sentence to support the statement, i.e. Maffiuletti et al. 2010 (already in the list of References) as well as Orssatto et al. 2020 and Lomborg et al. 2022

Orssatto LBR, Bezerra ES, Schoenfeld BJ, Diefenthaeler F. Lean, fast and strong: Determinants of functional performance in the elderly. Clin Biomech (Bristol, Avon) 78, 105073, 2020

Lomborg SD, Dalgas U, Hvid LG. The importance of neuromuscular rate of force development for physical function in aging and common neurodegenerative disorders - a systematic review. J Musculoskelet Neuronal Interact 22, 562-586, 2022

We thank the reviewer for the two points above. The sentence has been modified with "has been considered to be", and the references have been added (lines 104-105 of revised manuscript).

(7) Line 89

Suggest to add 'narrative', cf. above Comment 1:

"... this narrative review ..."

The word "narrative" has been added here and in all instances where we refer to the present review.

(8) Line 85

To improve the English syntax, suggest to replace 'work out' with 'address':

"... To address the specific mechanisms underlying the decline in explosive ..."

The sentence has been modified as suggested (line 108 of the revised manuscript).

(9) Line 99

suggest to insert 'physiological':

"... the impact on its physiological determinants ..."

The sentence has been modified with "neuromechanical" (line 122 of the revised manuscript).

(10) Line 105

Suggest to inform the Reader that this equation quantifies the production of impulse

"... to the following relationship describing the production of impulse: ..."

This section has been shortened, and the suggestion has been included (line 129 of the revised manuscript).

(11) Lines 115-116

To facilitate the concept of impulse to the Reader, suggest to cite Rodríguez-Rosell et al. 2018 (their Fig.4):

"... to produce the greatest impulse, the highest amount of force in the minimum time must be developed (Rodríguez-Rosell et al. 2018, their Fig.4) ..."

Reference to the work of Rodriguez-Rosell has been added. In addition, we have added reference to the work of Aagaard et al. (2002; lines 135-136 of the revised manuscript).

(12) Line 140

To be more consistent, suggest to insert 'produced during':

"... the impulse produced during the first 250 ms ..."

The text has been modified as suggested (lines 159-160 of the revised manuscript).

(13) Line 159

suggest to add 'postural':

"... in elderly, indices of postural balance correlated with RFD ..."

Similar relationships also have been observed in young adults (i.e. Jakobsen et al. 2011), which might deserve mentioning.

Jakobsen MD, Sundstrup E, Krstrup P, Aagaard P. The effect of recreational soccer training and running on postural balance in untrained men. *Eur. J. Appl. Physiol.* 111, 521-530, 2011

The text has been modified, and reference to the work of Jakobsen has been added (lines 178-181 of the revised manuscript).

(14) Lines 171-172

The use of RFD testing to guide a safe return to sport following musculoskeletal injury is an emerging yet important aspect in athlete rehabilitation, which is only rarely addressed in the literature. Consequently, consider also citing Jordan et al. 2023 to support/elaborate on this approach.

Jordan MJ, Aagaard P, Bishop C, McLean Z, Morris N, Boon-van Mossel N, Pasanen K, da Silva Torres R, Herzog W. Explosive strength and stretch-shortening-cycle capacity during ACL rehabilitation - Mechanical biomarkers for return to sport and performance readiness. *ASPETAR Sports Med J* 12, 324-331, 2023

We thank the reviewer for this suggestion. We indeed consider RFD a pivotal measure to assess in athletes for returning to play or to sports. We have added reference to the work of Jordan (2015; 2023; lines 190-192 of the revised manuscript).

(15) Line 223

I fail to see the relevance/information of this partial derivative of power relative to RFD, as it doesn't explain any features of the right-most graph (Power-time curve) shown in Figure 2.

I suggest the Authors to instead use the equation in line 221 to derive/develop the analytical relationship for the rate of power development, i.e. $dW[\dot{ }]/dt$. This relationship may be considered more interesting/relevant as it will describe the shape (slope) of the power-time curve shown in Figure 2, thus extending and explaining previous observations of training-induced improvements in the rate of power development reported in the literature (Jakobsen et al. 2012), which might

also be relevant to discuss at this point of the text.

Jakobsen MD, Sundstrup E, Randers MB, Kjær M, Andersen LL, Krstrup P, Aagaard P. The effect of strength training, recreational soccer and running exercise on stretch-shortening cycle muscle performance during countermovement jumping. *Hum Mov Sci* 31, 970-986, 2012

We thank the reviewer for the constructive comment. We have now added the formula for the rate of power development as the time derivative of power, and removed the partial derivative of power relative to RFD. We have also provided in Figure 2C the plot of power relative to time and RFD, and its projection on the power-time plane, to highlight how power in the simulation we used changes as function of an initial set RFD (lines 239-248 of the revised manuscript).

(17) Lines 234-236

It is stated that

"... ii) in dynamic movements, RFD (from a muscle or a muscle group) is affected by muscle-length and contraction-velocity dependent mechanisms (Hahn et al., 2017; Lieber & Ward, 2011) ..."

In addition, RFD is also affected by acute/instantaneous changes in fascicle pennation angle (shifts in gearing ratio) (Van Hooren et al. 2024), which perhaps should be briefly mentioned at this point in the text.

Van Hooren B, Aagaard P, Monte A, Blazevich AJ. The role of pennation angle and architectural gearing to rate of force development in dynamic and isometric muscle contractions. *Scand. J. Med. Sci. Sports* 34, e14639, 2024

We thank the reviewer for pointing this out. We have changed this section of the paper (lines 255-258 of the revised manuscript) to earlier introduce the dependency of RFD on muscle fascicle velocity and gear ratio. In addition, we have added to the manuscript the reference to the work of Van Hooren et al., 2024. Reference to the work of Van Hooren et al., 2024 has also been added in line 308 of the revised manuscript, where the neuromechanical determinants of explosive force in isometric and dynamic conditions are briefly reported.

(18) Lines 254

suggest to elaborate / be more concise:

"... high MU firing rates ..."

We have now modified the text to be more concise (lines 273-274 of the revised manuscript).

(20) Line 259

'increase' seems to be missing:

"... MUs to maximise their initial firing rate, and in turn increase RFD (Aoki et al., 2002 ..."

The text has been modified as suggested (line 277).

(21) Line 274

Suggest to change "... did not influence ..." into "... may not influence ..." since MU synchronization may not be very clearly distinguished by current HDEMG analysis algorithms.

The text has been modified as suggested (lines 292-294 of the revised manuscript). We have also added reference to the work of de la Rocha et al. (2007), showing that MUs synchronization may increase independent of common synaptic input only from higher MUs discharge rate.

(22) Line 280

Should say "... Bojsen-Møller et al. ..."

The surname has been changed throughout the text.

(23) Line 281

Suggest to also cite seminal study by Lichtwark and coworkers 2016.

Mayfield DL, Cresswell AG, Lichtwark GA. Effects of series elastic compliance on muscle force summation and the rate of force rise. J Exp Biol 219, 3261-3270, 2016

The reference to the work of Mayfield has been added (line 297 of the revised manuscript).

(24) Lines 287-291

In terms of the influence of muscle architectural gearing on RFD, suggest to also cite recent paper by Van Hooren et al. 2024.

Van Hooren B, Aagaard P, Monte A, Blazevich AJ. The role of pennation angle and architectural gearing to rate of force development in dynamic and isometric muscle contractions. Scand. J. Med. Sci. Sports 34, e14639, 2024

The work is now cited (line 308 of the revised manuscript).

(25) Lines 309-310

It is stated that "... Such discrepancy between eccentric and concentric contractions comes from neural inhibition of force production in the former modality ..."

To support this notion, suggest to cite reference(s) that have addressed the presence of neural inhibition during maximal eccentric muscle actions, i.e. Aagaard 2018.

Aagaard P. Spinal and supraspinal control of motor function during maximal eccentric muscle contraction: Effects of resistance training. J Sport Health Sci 7, 282-293, 2018

We thank the author for this observation. We have now cited the seminal reviews of Aagaard et al. (2018) and of Duchateau & Enoka (2016) on this topic (lines 323-324 of the revised manuscript).

(26) Line 319

Suggest to exchange 'needed' with 'relevant':

"... However, when relevant, reference will be made to dynamic explosive movements ..."

The text was changed as suggested.

(27) Lines 350-373

Suggest to delete paragraph listed in lines 350-373, to reduce the (somewhat excessive) length of the manuscript.

We agree with the reviewer that information in the last part of this section may be less relevant. We kept the paragraph indicated by the reviewer as we deem important to specify the reason why we did not consider in the present review findings from the important studies on reduced physical activity and long-term orthopedic limitations. Instead, we removed the last paragraph of this section, because it was related to paradigms of muscle unloading used in animals (lines 360-372 of the original manuscript).

(28) Line 418

Suggest to rephrase to improve the English syntax

"... applying a logarithmic equation

The text has been changed as suggested (line 412 of the revised manuscript).

(29) Line 454

Figure 3. For each data type, single data points represent a single study? Please make this more clear to the Reader.

The description of the figure has been modified to include that each data point represents data from each study (lines 449-451 of the revised manuscript). Also, the figure legend has been shortened, and we refer to the main text to explain details of the fitting procedures.

(30) Line 475

Should say "... IN EXPLOSIVE STRENGTH WITH DISUSE ..." ?

The text has been changed as suggested (line 466 of the revised manuscript). We also want to acknowledge that following the comments of the first reviewer, this section has been considerably modified. We have now reported the neural mechanisms by which MUs recruitment threshold and firing frequency may change after muscle unloading. We considered alterations in net excitatory drive to motoneurons (mainly corticospinal and reticulospinal pathways) and motoneurons excitability (altered neuromodulatory input and neurophysiological properties). In this way, the reader has a complete overview of the mechanistic reasons why MUs recruitment threshold and firing rate change, and why explosive strength should be more affected than maximal strength.

(31) Line 476

Exchange 'of' with 'in':

"... in Fmax ..."

The text has been changed as suggested.

(32) Line 477

To assist the Reader, suggest to add '(cf. Figure 3A)':

"... atrophy (cf. Figure 3A), as evident ..."

The text has been changed as suggested.

(33) Line 506

To improve consistency, suggest to add 'caused by changes in'

" ... or caused by changes in intrinsic motoneuronal properties ..."

The text is now changed given comments of the first reviewer.

(34) Line 507

Suggest to insert 'synaptic':

"... equal synaptic input to motoneurons ..."

The text is now changed given comments of the first reviewer.

(35) Line 566

Suggest to add 'however'

"... production, however at the expense ..."

The text is now changed given comments of the first reviewer.

(36) Line 624

Suggest to add 'level':

"... same excitation level. Combined ..."

The text is now changed given comments of the first reviewer.

(37) Line 630

To be more concise, suggest to add 'reduced':

"... found to decline, e.g., 18% reduced already after 14 days ..."

The text is now changed given comments of the first reviewer.

(38) Lines 631-632

It is stated that "... with longer intervention durations typically corresponding to a greater decrease (De Boer et al., 2007; Kubo et al., 2000; Kubo et al., 2004; Reeves et al., 2005; Roffino et al., 2021) ..."

How much decrease (% decline) in how many days? Please include this information in the text.

The decline found in the specific studies is now reported (lines 647-650 of the revised manuscript).

(39) Line 633

Suggest to use the pluralis form:

"... such declines in tendon stiffness ..."

The text is now changed given comments of the first reviewer.

(40) Lines 636-677

Suggest to omit this entire section (lines 636-677), to reduce the (very long) length of the review.

Given in our opinion the importance of some of the points in this section, we opted for keeping it in. However, this section has been considerably shortened. Also, given the additions required by the comments of the first reviewer, we have removed details of the Hill-type force model, and included them in the Supplementary Material (Section S4).

(41) Line 692

At the end of this section (Future directions), the Authors should consider including 2-4 sentences addressing how an increased knowledge of the factors mentioned/discussed in the present review may be used to establish more effective countermeasures (exercise paradigms, NMES, passive/active BFR, etc) to minimize the detrimental effect of short-term disuse on RFD, to thereby prevent/attenuate impairments in the functional performance of athletes, elderly/old adults and clinical patients, respectively.

Response. The text has been added at the end of the section “Future Directions” (lines 756-764 of the revised manuscript).

Dear Dr Ruggiero,

Re: JP-TR-2024-285667R1 "Neuromuscular mechanisms for the fast decline in rate of force development with muscle disuse - a narrative review." by Luca Ruggiero and Markus Gruber

We are pleased to tell you that your paper has been accepted for publication in The Journal of Physiology.

IMPORTANT

Two things to bring to your attention:

(1) Your paper contains Supporting Information of a type that we no longer publish. Please see our Supporting Information Guidelines.

To address this, we will instead move this information to an Appendix (within the article file). Please can you confirm that this meets with your approval.

(2) Abstract figure - we realise that there has been a problem with access to BioRender and providing a high resolution version. Diana at the Editorial Office will be in touch separately about this.

Authors should note that it is too late at this point to offer corrections prior to proofing. Major corrections at proof stage, such as changes to figures, will be referred to the Editors for approval before they can be incorporated. Only minor changes, such as to style and consistency, should be made at proof stage. Changes that need to be made after proof stage will usually require a formal correction notice.

Yours sincerely,

Laura Bennet
Senior Editor
The Journal of Physiology

P.S. - You can help your research get the attention it deserves! Check out Wiley's free Promotion Guide for best-practice recommendations for promoting your work at www.wileyauthors.com/eoo/guide. You can learn more about Wiley Editing Services which offers professional video, design, and writing services to create shareable video abstracts, infographics, conference posters, lay summaries, and research news stories for your research at www.wileyauthors.com/eoo/promotion.

IMPORTANT NOTICE ABOUT OPEN ACCESS: To assist authors whose funding agencies mandate public access to published research findings sooner than 12 months after publication, The Journal of Physiology allows authors to pay an Open Access (OA) fee to have their papers made freely available immediately on publication.

You can check if your funder or institution has a Wiley Open Access Account here: <https://authorservices.wiley.com/author-resources/Journal-Authors/licensing-and-open-access/open-access/author-compliance-tool.html>.

EDITOR COMMENTS

Reviewing Editor:

The authors are commended for conducting a thorough and valuable review. The revised paper makes a significant contribution to the targeted area of research (disuse effects on rapid force capacity) and offers deeper insights into key physiological mechanisms relevant to this topic.

REFEREE COMMENTS

Referee #1:

No further comments.

Referee #2:

The Authors are congratulated on performing a constructive and extensive review. The revised paper contributes with significant impact on the targeted area of research (disuse effects on rapid force capacity) while providing expanded insight(s) into important physiological mechanisms of importance to this theme.

1st Confidential Review

11-Sep-2024